# Diffusion Controller: Framework, Algorithms and Parameterization

**Tong Yang** [1 2]   **Moonkyung Ryu** [2]   **Chih-Wei Hsu** [2]   **Guy Tennenholtz** [2]   **Yuejie Chi** [3]   **Craig Boutilier** [2]   **Bo Dai** [4]

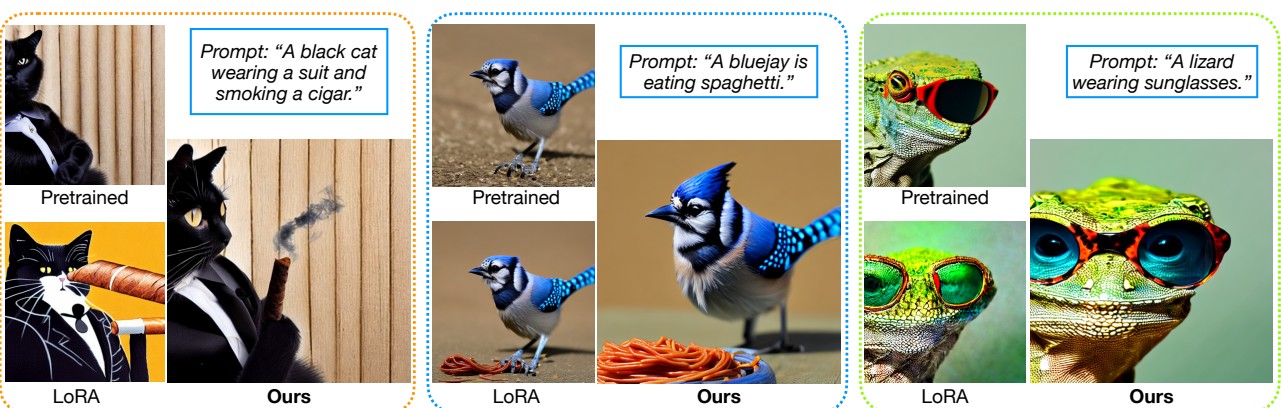

**Figure 1:** Qualitative text-to-image comparisons of pretrained Stable Diffusion v1.4, LoRA finetuning, and ours.

## Abstract

Controllable diffusion generation often relies on various heuristics that are seemingly disconnected without a unified understanding. We bridge this gap with Diffusion Controller (DiffCon), a unified control-theoretic view that casts reverse diffusion sampling as state-only stochastic control within (generalized) linearly-solvable Markov Decision Processes (LS-MDPs). Under this framework, control acts by reweighting the pretrained reverse-time transition kernels, balancing terminal objectives against an $f$-divergence cost. From the resulting optimality conditions, we derive practical reinforcement learning methods for diffusion fine-tuning: (i) $f$-divergence-regularized policy-gradient updates, including a PPO-style rule, and (ii) a regularizer-determined reward-weighted regression objective with a minimizer-preservation guarantee under the Kullback–Leibler (KL) divergence. The LS-MDP framework further implies a principled model form: the optimal score decomposes into a fixed pretrained baseline plus a lightweight control correction, motivating a side-

network parameterization conditioned on exposed intermediate denoising outputs, enabling effective *gray-box adaptation* with a frozen backbone. Experiments on Stable Diffusion v1.4 across supervised and reward-driven finetuning show consistent gains in preference-alignment win rates and improved quality–efficiency trade-offs versus gray-box baselines and even the parameter-efficient white-box adapter LoRA.

## 1. Introduction

Recent advances in diffusion models have enabled high-fidelity image synthesis, becoming a default choice for text-to-image generation (Ho et al., 2020; Song & Ermon, 2019), e.g., Stable Diffusion 3 (Esser et al., 2024), Flux.1 (Labs et al., 2025), Emu3 (Wang et al., 2024). Yet, controllable generation—steering samples to satisfy user intent, constraints, or downstream objectives—remains challenging, as stronger steering typically requires departing further from the pretrained model and can degrade sample quality (Zhang et al., 2023). In practice, control is achieved through a mix of inference-time mechanisms (e.g., classifier/classifier-free guidance and related guidance variants (Dhariwal & Nichol, 2021; Ho & Salimans, 2022)) and training-time adaptation (e.g., personalization or domain adaptation via paired supervision and parameter-efficient adapters (Ruiz et al., 2023; Zhang et al., 2023)). More recently, a growing line of work fine-tunes diffusion models using human feedback or learned rewards, aiming to optimize terminal objectives such

[1]Carnegie Mellon University, Pittsburgh, Pennsylvania, USA [2]Google Research [3]Yale University, New Haven, Connecticut, USA [4]Google DeepMind. Correspondence to: Tong Yang <tongyang@andrew.cmu.edu>.

*Proceedings of the $43^{rd}$ International Conference on Machine Learning*, Seoul, South Korea. PMLR 306, 2026. Copyright 2026 by the author(s).

as alignment, aesthetics, or task scores (Lee et al., 2023; Fan et al., 2023). Despite strong empirical progress, these approaches are often presented as a patchwork of objectives and disconnected heuristics, and the field has largely lacked a simple, principled and unified framework to model and analyze the control of these powerful generative processes.

## 1.1. Our contribution

To address this gap, we aim to develop a control-theoretic view of diffusion sampling that provides a common language for interpreting a range of diffusion finetuning approaches. To this end, we view diffusion finetuning as optimal control of the Markov process over intermediate noisy states along the denoising trajectory—induced by the reverse-time sampler. Under this lens, controllable diffusion generation amounts to learning a controlled reverse-time transition kernel of diffusion that steers the induced terminal distribution toward a target objective.

Under this controlled-sampling view, diffusion finetuning becomes a *state-only* stochastic control problem that fits naturally into the *linearly-solvable MDP* (LS-MDP) framework (Todorov, 2006). In contrast to standard MDP formulations that introduce explicit actions (Uehara et al., 2024a), our LS-MDP–induced diffusion controller (DiffCon) directly modulates the reverse-time dynamics by reweighting the passive (uncontrolled) pretrained transitions. We penalize this modulation with a general $f$-divergence control cost (recovering the classic KL-regularized LS-MDP (Todorov, 2006) as a special case), which provides a principled trade-off between reward-driven steering and staying close to the pretrained model for preserving stability and sample quality.

We next turn the LS-MDP characterization into implementable diffusion finetuning algorithms by realizing the optimality conditions with a score-parameterized diffusion reverse process, rather than directly learning the implied optimal reverse kernels that are hard to sample from. This translation yields practical objectives in the reinforcement learning finetuning (RLFT) setting where no target samples are available and feedback is provided only by a terminal reward model on the final image. Specifically, we derive two RL finetuning updates: **i)**, $f$-divergence-regularized policy-gradient methods (Williams, 1992) with standard extensions such as PPO (Schulman et al., 2017); and **ii)**, regularizer-determined reward-weighted regression objectives with a minimizer-preservation guarantee (under KL, matching the relevant LS-MDP optimal marginal).

Finally, beyond offering a unified objective template, the LS-MDP view also sheds light on the optimal model form: the optimal controlled reverse process is a regularized perturbation of the pretrained diffusion model (aka the backbone), so the learned score naturally decomposes into a fixed pretrained baseline plus a small control correction.

This motivates our DiffCon parameterization that freezes the backbone and learns a lightweight controller network conditioned on exposed intermediate denoising outputs (e.g., per-step noise predictions or the implied reverse mean), making it naturally compatible with *gray-box* settings where the backbone architecture is unknown (Poole et al., 2022). Our parameterization steers the backbone model with a small and interpretable module that preserves pretrained stability yet enables strong and tunable control.

Our experiments demonstrate the effectiveness of our proposed algorithms and parameterization. Across both SFT and RLFT settings, the resulting objectives significantly improve win rate over the pretrained model. Moreover, the LS-MDP-motivated score reparameterization delivers a stronger quality–efficiency trade-off—outperforming *gray-box* baselines and the *white-box* adapter LoRA.

## 1.2. Related work

**Reinforcement learning for diffusion models.** Existing RL finetuning for diffusion models treats denoising as an MDP and optimizes terminal rewards via policy gradients (e.g., DDPO (Black et al., 2023), DPOK (Fan et al., 2023)), differentiable-reward backpropagation (e.g., DRaFT (Clark et al., 2023)), or reward-weighted likelihood/score matching (Lee et al., 2023; Fan et al., 2023), see Uehara et al. (2024a) for a review. Several works align diffusion models from pairwise preferences using DPO-style objectives (Wallace et al., 2024; Yang et al., 2024a; Kim et al., 2025; Croitoru et al., 2025), and others mitigate delayed supervision by constructing dense or step-level rewards along the denoising trajectory (Yang et al., 2024b). Tang & Zhou (2024) further connects *continuous-time* diffusion finetuning to entropy/$f$-divergence regularized stochastic control, with follow-ups casting diffusion finetuning explicitly as entropy-regularized control (Uehara et al., 2024b; Han et al., 2024).

**Score-function parameterization and modular control.** A common diffusion finetuning practice keeps a strong pretrained backbone fixed and steers generation with lightweight adapters (e.g., LoRA (Hu et al., 2022), or Diff-Fit (Xie et al., 2023), which tunes only bias terms plus a small set of added layerwise scaling factors) or side networks such as ControlNet (Zhang et al., 2023), T2I-Adapter (Mou et al., 2024), IP-Adapter (Ye et al., 2023), Gligen (Li et al., 2023) that add score adjustments (akin to guidance). In contrast, our LS-MDP analysis yields a principled pretrained+controller decomposition of the optimal score that supports gray-box access, with the control term implemented as a side module parameterized via random Fourier features (Rahimi & Recht, 2007; Tancik et al., 2020).

**Notation.** Let $\mathbb{R}, \mathbb{N}$ be the set of real numbers and natural numbers. Let $\mathbb{R}^d$ be the $d$-dimensional real vector space, and $\|x\|_2$ represent the Euclidean norm of $x$. The Gaussian distribution with mean $\mu$ and covariance matrix $\Sigma$ is denoted by $\mathcal{N}(x|\mu, \Sigma)$, and $\mathcal{U}(\mathcal{A})$ stands for the uniform distribution over set $\mathcal{A}$. Define $[T] := \{1, \cdots, T\}$. Last but not least, $\mathcal{O}(\cdot)$ and $\mathcal{O}_p(\cdot)$ denote the standard asymptotic big-O notation and the big-O in probability notation, respectively.

## 2. Preliminaries

**Linearly-Solvable MDP (LS-MDP) (Todorov, 2006).** An LS-MDP is a stochastic control problem in which the controller acts directly on the transition kernel: at each time step, it selects a next-state distribution that stays close to a given *passive* (uncontrolled) dynamic, measured by the Kullback-Leibler (KL) divergence. Concretely, let $s_t \in \mathcal{S}$ be the state at time $t$ from the state space $\mathcal{S}$, and $\mathbb{P}_t(s_{t+1} \mid s_t)$ denote the passive transition kernel at time $t$. The time-dependent *control* $u_t : \mathcal{S} \times \mathcal{S} \to \mathbb{R}$ exponentially tilts the passive dynamic as:

$$\mathbb{P}_{u,t}(s_{t+1}|s_t) = \mathbb{P}_t(s_{t+1}|s_t) \exp(u_t(s_{t+1}, s_t)),$$

$$\text{s.t.} \int_{\mathcal{S}} \mathbb{P}_{u,t}(s_{t+1}|s_t) ds_{t+1} = 1, \quad \forall t \in [T-1], \quad (1)$$

where $T$ is the time horizon. Given an instantaneous reward $r_t : \mathcal{S} \to \mathbb{R}$, the (regularized) value recursion is

$$V_{u,t}(s_t) := r_t(s_t) + \mathbb{E}_{s_{t+1} \sim \mathbb{P}_{u,t}(\cdot|s_t)} [V_{u,t+1}(s_{t+1})]$$
$$-\tau D_{\text{KL}}(\mathbb{P}_{u,t}(\cdot|s_t) \| \mathbb{P}_t(\cdot|s_t)), \quad (2)$$

where $D_{\text{KL}}(p \| q) := \int p(x) \log \frac{p(x)}{q(x)} dx$ is the KL divergence, and $\tau \geqslant 0$ is the regularization coefficient.

The KL-regularized formulation (2) has also been generalized by replacing the KL control cost with more general divergence regularizers, such as Rényi-regularized LS-control (Dvijotham & Todorov, 2011) or Tsallis entropy (Hashizume et al., 2024).

**Diffusion models (Ho et al., 2020).** Diffusion models generate data by learning to iteratively transform a simple noise distribution into a target data distribution. We describe the setting of *conditional generation*, where clean images follow the distribution $x_T \sim p_{\text{data}}(\cdot \mid c)$ conditioned on $c \sim p_c$ (e.g., class labels or text), with condition support $\mathcal{C}$.[1] When $c = \emptyset$, this reduces to unconditional generation. Let $\{\beta_t \in (0,1)\}_{t=1}^{T-1}$ be the variance schedule, and define

$$\alpha_t = 1 - \beta_t, \quad \overline{\alpha}_t = \prod_{i=t}^{T-1} \alpha_i. \quad (3)$$

We now describe the forward process and the reverse process used in diffusion models.

- **Forward process.** The forward (noising) process is a *fixed* Markov chain that gradually corrupts a clean image into (approximately) standard Gaussian noise. Starting from $x_T \sim p_{\text{data}}(\cdot \mid c)$, for any $t \in [T-1]$,

$$x_t = \sqrt{\alpha_t} x_{t+1} + \sqrt{\beta_t} \xi_t = \sqrt{\overline{\alpha}_t} x_T + \sqrt{1 - \overline{\alpha}_t} \xi, \quad (4)$$

where $\xi_t, \xi \sim \mathcal{N}(0, \mathbf{I}_d)$. In particular, $x_1$ is approximately standard Gaussian. We use score matching to learn the score function $\epsilon(\cdot, c, t)$,[2] which is parameterized by neural networks as $\epsilon_\theta(\cdot, c, t)$, using the following noise-prediction regression objective:

$$\min_\theta \mathbb{E}_{c, x_T, t, \xi} \left[ \frac{1}{2} \left\| \xi - \epsilon_\theta \left( \sqrt{\overline{\alpha}_t} x_T + \sqrt{1 - \overline{\alpha}_t} \xi, c, t \right) \right\|_2^2 \right],$$

where the expectation is taken over $c \sim p_c$, $x_T \sim p_{\text{data}}(\cdot|c)$, $t \sim \mathcal{U}([T-1])$, and $\xi \sim \mathcal{N}(0, \mathbf{I}_d)$.

- **Reverse process.** At sampling time, diffusion models iteratively denoise a noise sample $x_1 \sim \mathcal{N}(0, \mathbf{I}_d)$ to $x_T$ through the Gaussian reverse-time transitions:

$$x_{t+1} \sim p_t(\cdot|x_t, c) := \mathcal{N}(\cdot|\mu(x_t, c, t), \widetilde{\beta}_t \mathbf{I}_d) \quad (5)$$

with mean

$$\mu(x_t, c, t) = \frac{1}{\sqrt{\alpha_t}} x_t - \frac{\beta_t}{\sqrt{\alpha_t(1 - \overline{\alpha}_t)}} \epsilon_\theta(x_t, c, t)$$

and reverse-time variance $\widetilde{\beta}_t := \frac{\beta_t(1 - \overline{\alpha}_{t+1})}{1 - \overline{\alpha}_t}$. For unconditional generation, we directly set $\epsilon_\theta(x_t, t) = \epsilon_\theta(x_t, \emptyset, t)$ to generate the image following the reverse process.

For conditional generation, we use the standard *classifier-free guidance (CFG)* framework (Ho & Salimans, 2022): during training we drop the condition with probability $p_{\text{drop}} \in (0, 1)$, and at sampling we combine conditional and unconditional predictions as

$$\hat{\epsilon}_\theta(x_t, c, t) = (1 + \lambda_{\text{CFG}})\epsilon_\theta(x_t, c, t) - \lambda_{\text{CFG}}\epsilon_\theta(x_t, \emptyset, t),$$

where $\lambda_{\text{CFG}} \geqslant 1$ controls guidance strength.

**Access levels for diffusion finetuning.** We categorize diffusion fine-tuning by access to the pretrained backbone. In the *white-box* setting, the backbone can be modified (e.g., full finetuning or inserting adapters such as LoRA (Hu et al., 2022)). In many real-world applications, however, the backbone may be sealed for proprietary or safety reasons, while exposing limited interfaces (e.g., per-timestep noise predictions or denoising trajectories). This *gray-box* setting motivates finetuning with add-on modules that harvest on these interfaces while keeping the backbone intact (Poole et al., 2022; Yang et al., 2024c; Yu et al., 2023b; Zhu et al., 2025; Yu et al., 2023a).

---

[1]Our indexing is reversed from the convention used in Ho et al. (2020) to align with common RL notation: $x_T$ is the clean image and $x_1$ is pure noise.

[2]To be precise, this is the noise-prediction function, which can be mapped equivalently from the score function (Song et al., 2020).

## 3. Diffusion Controller through LS-MDP

We are interested in finetuning a pretrained diffusion model to generate images from a new distribution (e.g., conditional generation, human-preference alignment, domain adaptation, etc.). In this section, we formulate diffusion finetuning through the lens of LS-MDP control, which provides a simpler and unified framework compared to prior works using standard MDPs (Black et al., 2023; Fan et al., 2023). We call the proposed framework *Diffusion Controller (DiffCon)*: a controllable diffusion reverse process that steers intermediate states to match a target distribution. Building on this formulation, we further derive the corresponding RL finetuning algorithms.

### 3.1. Framework

Prior works (e.g., DDPO (Black et al., 2023), DPOK (Fan et al., 2023)) cast the reverse diffusion chain as a standard MDP by taking the state $s_t = (x_t, c, t)$ and treating the next sample as the action $a_t = x_{t+1}$, with a deterministic environment and a policy given by the reverse transition from $x_t$ to $x_{t+1}$. However, this "action" is simply (part of) the next state, so the only real degree of freedom is how we shape the transition kernel itself. This motivates our LS-MDP characterization, which directly controls the reverse transition (regularized to stay close to the pretrained dynamics) rather than introducing a separate action variable, greatly simplifying the presentation.

Let $\epsilon_0(x_t, c, t)$ be the pretrained score function, whose reverse process is given by $x_1 \sim p_{0,1} := \mathcal{N}(\cdot|0, \mathbf{I}_d)$, and for all $t \in [T-1]$:

$$x_{t+1} \sim p_{0,t}(\cdot|x_t, c) := \mathcal{N}(\cdot|\mu_0(x_t, c, t), \widetilde{\beta}_t \mathbf{I}_d) \quad (6)$$

same as in (5), with mean

$$\mu_0(x_t, c, t) := \frac{1}{\sqrt{\alpha_t}} x_t - \frac{\beta_t}{\sqrt{\alpha_t(1-\overline{\alpha}_t)}} \epsilon_0(x_t, c, t). \quad (7)$$

We use the generalized LS-MDP framework (c.f. (1), (2)) to characterize the target distribution of finetuning. We define the controllable transition kernels induced by *control* $u_t : \mathbb{R}^d \times \mathbb{R}^d \times \mathcal{C} \to \mathbb{R}$ as

$$\mathbb{P}_{u,t}(x_{t+1}|x_t, c) = p_{0,t}(x_{t+1}|x_t, c) \exp(u_t(x_{t+1}, x_t, c)),$$

$$\text{s.t.} \int_{\mathbb{R}^d} \mathbb{P}_{u,t}(x_{t+1}|x_t, c) dx_{t+1} = 1 \quad (8)$$

for all $c \in \mathcal{C}$ and $t = 0, \cdots, T-1$. Here we introduce $x_0 := \emptyset$ for controlling the initial distribution $p_{0,0}(x_1|x_0, c) := p_{0,1}(x_1)$ of $x_1$ with $u_0$. Given a reward function $r_t(x_t, c)$, we let $u^\star = \{u_t^\star\}_{t=0}^{T-1}$ denote the optimal control that solves

the following value maximization problem:

$$\max_{u_t(\cdot, \cdot, c)} V_{u,t}(x_t, c)$$
$$:= r_t(x_t, c) + \mathbb{E}_{x_{t+1} \sim \mathbb{P}_{u,t}(\cdot|x_t, c)}[V_{u,t+1}(x_{t+1}, c)]$$
$$- \tau D_f(\mathbb{P}_{u,t}(\cdot|x_t, c) \| p_{0,t}(\cdot|x_t, c)) \quad (9)$$

for $c \in \mathcal{C}$ and $t = 0, \cdots, T-1$ with terminal value function $V_T(x_T, c) := r_T(x_T, c)$ and regularization strength $\tau \geqslant 0$, where $D_f$ is an $f$-divergence (Csiszár, 1963): $D_f(p \| q) = \int f\left(\frac{p(x)}{q(x)}\right) q(x) dx$ for some convex $f$ with $f(1) = 0$. When $D_f = D_{\text{KL}}$, (9) recovers the classical KL-regularized LS-MDP (2).

**Our goal.** We aim to learn the optimal transition kernel $\mathbb{P}_{u^\star,t}(\cdot|x_t, c)$ that leads to our *target distribution* $p_{u^\star}(\cdot|c)$ (given any $c \in \mathcal{C}$)—the terminal sample distribution of $x_T$ under the optimal control $u^\star$.

However, $\mathbb{P}_{u^\star,t}(\cdot|x_t, c)$ characterized by the LS-MDP optimality conditions is generally intractable to sample from (e.g., in the KL-regularized case, it takes an energy-based form, see (51) in the Appendix), so naively sampling $x_{t+1} \sim \mathbb{P}_{u^\star,t}(\cdot|x_t, c)$ requires expensive inner-loop estimation at each step. To obtain an efficient ancestral sampler, we therefore learn a tractable diffusion reverse process whose induced transitions lead to the terminal distribution $p_{u^\star}$. Namely, we target for a *score function* $\epsilon^\star$ that minimizes the following regression loss $\mathcal{L}_{\text{org}}(\epsilon; p_{u^\star})$:

$$\frac{1}{2} \mathbb{E}_{\substack{c,t,\xi \\ x_T \sim p_{u^\star}(\cdot|c)}} \| \xi - \epsilon(\sqrt{\overline{\alpha}_t} x_T + \sqrt{1-\overline{\alpha}_t} \xi, c, t) \|_2^2, \quad (10)$$

where $c \sim p_c$, $t \sim \mathcal{U}([T-1])$, and $\xi \sim \mathcal{N}(0, \mathbf{I}_d)$.

### 3.2. Reinforcement Learning Finetuning (RLFT)

When target samples from $p_{u^\star}(\cdot|c)$ are unavailable and only a reward model is given, we derive tractable RL-style updates from the LS-MDP optimality in (9).

Same as existing work (Black et al., 2023; Fan et al., 2023; Lee et al., 2023), we assume a terminal reward model is available: the reward scores only the fully denoised sample at the end of the reverse process, *i.e.*, in (9) we have

$$r_t(x_t, c) = \begin{cases} r(x_T, c), & t = T, \\ 0, & \text{otherwise.} \end{cases} \quad (11)$$

Various diffusion control algorithms can be derived under the LS-MDP framework. To give some examples, we first derive the policy gradient type methods, and then propose reward-weighted losses for RL finetuning.

#### 3.2.1. POLICY GRADIENT AND PPO FOR DIFFCON

We use $p_{\theta,t}(x_{t+1}|x_t, c)$ to denote the transition kernel with learnable parameters $\theta$. It is often parameterized as the

Gaussian (5) for ease of sampling:

$$p_{\theta,t}(x_{t+1}|x_t,c) = \mathcal{N}(x_{t+1}|\mu_\theta(x_t,c,t), \widetilde{\beta}_t \mathbf{I}_d) \quad (12)$$

with

$$\mu_\theta(x_t,c,t) := \frac{1}{\sqrt{\alpha_t}} x_t - \frac{\beta_t}{\sqrt{\alpha_t(1-\overline{\alpha}_t)}} \epsilon_\theta(x_t,c,t), \quad (13)$$

where $\epsilon_\theta(x_t,c,t)$ is the score function to be learned. Let $p_{\theta,0}(x_1|x_0,c) := p_{\theta,0}(x_1|c)$, and write $\mathbb{E}_{x_{t+1:t'}|p_\theta,x_t,c}[\cdot]$ as the shorthand for $\mathbb{E}_{x_{s+1}\sim p_{\theta,s+1}(\cdot|x_s,c)}[\cdot]$ for any $t+1 \leqslant s \leqslant t'-1$ $t \leqslant s \leqslant t'-1$ $t' \leqslant T$, and define the value function under $p_{\theta,t}(\cdot|c)$ as

$$V_{\theta,t}(x_t,c) := r_t(x_t,c) + \mathbb{E}_{x_{t+1}|p_\theta,x_t,c}[V_{\theta,t+1}(x_{t+1},c)] \\ - \tau D_f(p_{\theta,t}(\cdot|x_t,c) \| p_{0,t}(\cdot|x_t,c)). \quad (14)$$

Unfolding (14), under our terminal reward (11), we have

$$V_{\theta,t}(x_t,c) = \mathbb{E}_{x_{t+1:T}|p_\theta,x_t,c}\left[ r(x_T,c) - \tau \sum_{s=t}^{T-1} \frac{f(\zeta_{\theta,s+1})}{\zeta_{\theta,s+1}} \right], \quad (15)$$

where $\zeta_{\theta,s+1} := \frac{p_{\theta,s}(x_{s+1}|x_s,c)}{p_{0,s}(x_{s+1}|x_s,c)}$. We define the soft advantage function given $(x_t,c,t)$ as

$$A_{\theta,t+1} := V_{\theta,t+1}(x_{t+1},c) - \tau f'(\zeta_{\theta,t+1}) - V_{\theta,t}(x_t,c) \\ + \tau \mathbb{E}_{x_{t+1}|p_\theta,x_t,c}\left[ f'(\zeta_{\theta,t+1}) - \frac{f(\zeta_{\theta,t+1})}{\zeta_{\theta,t+1}} \right]. \quad (16)$$

Then $A_{\theta,t+1}$ satisfies $\mathbb{E}_{x_{t+1}|p_\theta,x_t,c}[A_{\theta,t+1}] = 0$. And a special case is when $D_f = D_{\mathrm{KL}}$, we have

$$A_{\theta,t+1} = V_{\theta,t+1}(x_{t+1},c) - \tau \log(\zeta_{\theta,t+1}) - V_{\theta,t}(x_t,c).$$

The following proposition gives the policy gradient expression for maximizing

$$J_\theta := \mathbb{E}_{\substack{c\sim p_c, \\ x_1\sim p_{\theta,1}(\cdot|c)}}[V_{\theta,1}(x_1,c)] = \mathbb{E}_{c\sim p_c}[V_{\theta,0}(x_0,c)]$$

under the generalized LS-MDP framework.

**Proposition 1 (policy gradient)** *Under our reward setting (11), the gradient of $J_\theta$ is given by*

$$\nabla_\theta J_\theta = \mathbb{E}_{\substack{c\sim p_c, \\ x_{1:T}|p_\theta,c}}\left[ \sum_{t=0}^{T-1} \nabla_\theta \log p_{\theta,t}(x_{t+1}|x_t,c) A_{\theta,t+1} \right].$$

The proof of Proposition 1 is given in Appendix B.1.

Further, utilizing (16), we also give a PPO update rule for DiffCon with clipping parameter $\delta \in (0,1)$:

$$\theta^{(k+1)} \leftarrow \theta^{(k)} + \eta \nabla_\theta \mathbb{E}_{\substack{c\sim p_c, \\ x_{1:T}|p_{\theta^{(k)}},c}}\left[ \sum_{t=0}^{T-1} \min \left\{ \right.\right.$$

$$\mathrm{clip}\left( \frac{p_{\theta,t}(x_{t+1}|x_t,c)}{p_{\theta^{(k)},t}(x_{t+1}|x_t,c)}, 1-\delta, 1+\delta \right) A_{t+1}^{(k)},$$

$$\left.\left. \frac{p_{\theta,t}(x_{t+1}|x_t,c)}{p_{\theta^{(k)},t}(x_{t+1}|x_t,c)} A_{t+1}^{(k)} \right\} \right]\Bigg|_{\theta=\theta^{(k)}}. \quad (17)$$

**Relationship to existing policy gradient methods for diffusion finetuning.** Prior RL fine-tuning works such as DDPO (Black et al., 2023) and DPOK (Fan et al., 2023) apply policy gradients to the MDP described at the beginning of Section 3.1, taking the policy to be our reverse transition kernel $p_{\theta,t}(x_{t+1}|x_t,c)$. DDPO considers the unregularized setting, and our policy gradient recovers theirs when $\tau = 0$. DPOK adds a KL regularizer toward the pretrained policy, and derives a policy gradient close to ours (but different in the regularization term) when $D_f = D_{\mathrm{KL}}$, see Appendix A for a detailed comparison.

### 3.2.2. REWARD-WEIGHTED LOSS FOR DIFFCON

Besides policy-gradient updates, we can obtain $\epsilon^\star$ by reformulating the intractable objective in (10) into a tractable reward-weighted surrogate that preserves the same minimizer $\epsilon^\star$ (Ma et al., 2025).

To do this, we first introduce the following $f$-divergence regularized optimization problem on trajectory distribution, where we let $P_0(\cdot|c)$ be the joint reverse trajectory distribution over $x_{1:T}$ under the pretrained model conditioned on $c$:

$$\max_{P(\cdot|c)} \mathbb{E}_{P(\cdot|c)}[r(x_T,c)] - \tau D_f(P(\cdot|c) \| P_0(\cdot|c)) \quad (18)$$

for all $c \in \mathcal{C}$. We let $\widetilde{P}^\star(\cdot|c)$ be the optimal solution of (18), and let $\widetilde{p}^\star(\cdot|c)$ be the marginal distribution of $x_T$ under $\widetilde{P}^\star(\cdot|c)$. Theorem 1 gives the tractable reward-weighted loss as the substitute of (10) under KL divergence, by exploiting the equivalence between (18) and (9) when $D_f = D_{\mathrm{KL}}$. Moreover, it also gives the reward weighted loss equivalent to $\mathcal{L}_{\mathrm{org}}(\cdot; \widetilde{p}^\star)$ under general $f$-divergence regularization.

**Theorem 1 (reward weighted loss)** *Under our reward setting (11), when $\tau > 0$, the reward-weighted loss*

$$\mathcal{L}_f(\epsilon) := \frac{1}{2} \mathbb{E}_{\substack{t\sim \mathcal{U}([T-1]),\xi\sim\mathcal{N}(0,\mathbf{I}_d), \\ c\sim p_c, x_T\sim p_{0,T}(\cdot|c)}}\left[ w_f(r(x_T,c),\tau) \right.$$

$$\left. \cdot \left\| \xi - \epsilon\left( \sqrt{\overline{\alpha}_t}x_T + \sqrt{1-\overline{\alpha}_t}\xi, c, t \right) \right\|_2^2 \right] \quad (19)$$

*with the weighting function*

$$w_f(r(x_T,c),\tau) = \max\left\{ 0, (f')^{-1}\left( \frac{r(x_T,c) - b_{f,\tau}(c)}{\tau} \right) \right\}$$

*has the same minimum as $\mathcal{L}_{\mathrm{org}}(\cdot; \widetilde{p}^\star)$, where $b_{f,\tau}: \mathcal{C} \to \mathbb{R}$ is a problem-dependent baseline function of condition $c$. Specially, when $D_f = D_{\mathrm{KL}}$, $\widetilde{p}^\star = p_{u^\star}$.*

The proof of Theorem 1 is given in Appendix B.2.

As special examples of Theorem 1, for $D_f = D_{\mathrm{KL}}$, we have the following exponential weighting function

$$w_{\mathrm{KL}}(r(x_T,c),\tau) := \exp\left( \frac{r(x_T,c)}{\tau} \right); \quad (20)$$

for $D_f = D_\alpha$ ($\alpha$-divergence), and the following polynomial weighting function

$$w_\alpha(r(x_T, c), \tau) := \left[ 1 + \frac{\alpha - 1}{\tau} \left( r(x_T, c) - b_{\alpha.\tau}(c) \right) \right]_+^{\frac{1}{\alpha - 1}} \tag{21}$$

for some baseline function $b_{\alpha, \tau} : \mathcal{C} \to \mathbb{R}$.

Note that for general $f$-divergence $D_f \neq D_{\mathrm{KL}}$, $\widetilde{p}^\star$ typically differs from $p_{u^\star}$, so there is generally no closed-form weighting $w_f(r(x_T, c), \tau)$ exactly equivalent to (10). Nevertheless, the general reward-weighted loss form (19) remains a meaningful surrogate—it corresponds to optimizing the score-matching objective under the $f$-regularized optimal marginal $\widetilde{p}^\star$, and our experiments demonstrate the effectiveness of both the polynomial weighting (21) and the exponential weighting (20).

**Relationship to existing RWL methods for diffusion finetuning.** Prior diffusion RLFT work has also explored reward-weighted objectives: Lee et al. (2023) and Fan et al. (2023, Algorithm 2) optimize a reward-weighted likelihood-style loss on model samples with the weighting being the reward $r(x_T, c)$ itself. Uehara et al. (2024a, Section 6.1) gives a reward-weighted MLE/score-matching objective with the exponential weighting same as our (20). See Appendix A for a more detailed discussion.

## 4. Diffusion Controller: Parameterization

The LS-MDP formulation implies that the optimal reverse dynamics is not an arbitrary diffusion model—it is the result of a regularized control that stays close to the pretrained reverse kernel (6). This suggests a design principle: rather than fully re-learning a high-capacity score $\epsilon_\theta$, we would like a parameterization that **(i)**, keeps the pretrained score $\epsilon_0$ as a strong baseline and isolates the control signal into a lightweight module, and **(ii)** is compatible with *gray-box* access, where the backbone may be hidden but we can still query intermediate denoising outputs (e.g., $\epsilon_0$ or the implied reverse mean $\mu_0$).

To make this concrete, we focus on the KL-regularized case. Under the assumption that the pretrained score function $\epsilon_0$ minimizes the pretrained loss:

$$\mathcal{L}_0(\epsilon) := \frac{1}{2} \mathbb{E}_{\substack{c, t, \xi, \\ x_T \sim p_0(\cdot | c)}} \left[ \left\| \xi - \epsilon \left( \sqrt{\overline{\alpha}_t} x_T + \sqrt{1 - \overline{\alpha}_t} \xi, c, t \right) \right\|_2^2 \right] \tag{22}$$

where in the expectation, $p_0$ is the pretraining data distribution, $c \sim p_c$, $t \sim \mathcal{U}([T-1])$, and $\xi \sim \mathcal{N}(0, \mathbf{I}_d)$, we next ask: *what structure does the LS-MDP optimality impose on the optimal score $\epsilon^\star$?*

We will introduce Proposition 2 that expresses $\epsilon^\star$ in terms of $\epsilon_0$ plus two control-dependent objects that arise from

reweighting the pretrained reverse transition $p_{0,t}(\cdot | x_t, c)$. Further, the specific Gaussian form of $p_{0,t}(\cdot | x_t, c)$ admits a convenient Fourier feature representation that is widely used in representation learning (Rahimi & Recht, 2007; Tancik et al., 2020), allowing us to express the two control-dependent objects as linear combinations of a shared Fourier-style basis function set $\phi_\omega(x_t, c, t)$ defined as

$$\phi_\omega(x_t, c, t) := \begin{pmatrix} \cos\left( \omega^\top \mu_0(x_t, c, t) / \sqrt{\widetilde{\beta}_t} \right) \\ \sin\left( \omega^\top \mu_0(x_t, c, t) / \sqrt{\widetilde{\beta}_t} \right) \end{pmatrix}. \tag{23}$$

Now we state the main proposition of this section, where $q_{0,t}(\cdot | x_t, c)$ in the error term is the true posterior transition kernel induced by the data distribution $p_0$ and the forward process (that the learned reverse transition $p_{0,t}(\cdot | x_t, c)$ tries to approximate):

**Proposition 2** *Assume $D_f = D_{\mathrm{KL}}$ in (9), $\epsilon_0$ minimizes the pretrained loss (22), and all $x_T$ satisfy $\|x_T\|_\infty \leqslant 1$. Then the minimum of (10) can be represented as:*

$$\epsilon^\star(x_t, c, t)$$
$$= (1 - z_t^\star(x_t, c)) \epsilon_0(x_t, c, t) - \frac{\sqrt{1 - \overline{\alpha}_t}}{\beta_t} \Big( z_t^\star(x_t, c) x_t$$
$$+ \sqrt{\alpha_t}(1 - z_t^\star(x_t, c)) h_t^\star(x_t, c) - \sqrt{\alpha_t} \delta_{0,t} \Big) \tag{24}$$

*with*

$$h_t^\star(x_t, c) = \frac{1}{M} \sum_{i=1}^{M} W^\star(\omega_i, c, t) \phi_{\omega_i}(x_t, c, t) + \delta_{h,t}, \tag{25}$$

$$\frac{z_t^\star(x_t, c)}{1 - z_t^\star(x_t, c)} = \frac{1}{M} \sum_{i=1}^{M} u^\star(\omega_i, c, t)^\top \phi_{\omega_i}(x_t, c, t) + \delta_{z,t} \tag{26}$$

*for some functions $W^\star : \mathbb{R}^d \times \mathcal{C} \times [T] \to \mathbb{R}^{d \times 2}$ and $u^\star : \mathbb{R}^d \times \mathcal{C} \times [T] \to \mathbb{R}^2$, where $\{\omega_i\}_{i=1}^{M} \overset{i.i.d.}{\sim} \mathcal{N}(\cdot | 0, \mathbf{I}_d)$, $M$ is the sampling number, and the error terms $\delta_{h,t}, \delta_{z,t}, \delta_{0,t} \in \mathbb{R}^d$ satisfy $\|\delta_{h,t}\|_2 = \mathcal{O}_p\left(\frac{1}{\sqrt{M}}\right)$, $\|\delta_{z,t}\|_2 = \mathcal{O}_p\left(\frac{1}{\sqrt{M}}\right)$, $\|\delta_{0,t}\|_2 = \mathcal{O}\left(\sqrt{\beta_t d} \mathrm{TV}\left(q_{0,t}(\cdot | x_t, c), p_{0,t}(\cdot | x_t, c)\right)\right)$.*

The proof of Proposition 2 and the hidden problem-dependent constants in $\mathcal{O}$ and $\mathcal{O}_p$ are given in Appendix B.3.

The main implication of (24) in Proposition 2 indicates the optimal score $\epsilon^\star$ can be represented as a combination of the pretrained score $\epsilon_0$ and two control-dependent objects $h_t^\star(x_t, c)$ and $z_t^\star(x_t, c)$. Further, (25) and (26) show both $h^\star$ and the logit of $z^\star$ can be approximated by a linear combination of shared basis features $\{\phi_{\omega_i}(x_t, c, t)\}_{i=1}^{M}$, where all $x_t$-dependence enters through $\mu_0(x_t, c, t)$ in $\phi_{\omega_i}(x_t, c, t)$. This is structurally analogous to cross-attention with $\mu_0(x_t, c, t)$ replacing $x_t$ as the input, where the condition and time $(c, t)$ produce weights that select and mix latent features extracted from $\mu_0$. This motivates our DiffCon parameterization of the score function in the following.

**DiffCon score parameterization.** Proposition 2 inspires us to use a "side network" (Zhang et al., 2020) $s_\theta = (z_{\theta_1}, h_{\theta_2})$ to combine with $\epsilon_0$ to parameterize $\epsilon_\theta$ as

$$\epsilon_\theta(x_t, c, t) = \epsilon_0(x_t, c, t) + \lambda_{\text{model}} s_\theta(\mu_0(x_t, c, t), c, t) \quad (27)$$

with $\mu_0(x_t, c, t)$ instead of $x_t$ as the input to $s_\theta$, where

$$s_\theta(x, c, t) := -z_{\theta_1}(x, c, t)\epsilon_0(x, c, t) - \frac{\sqrt{1 - \overline{\alpha_t}}}{\beta_t}$$
$$\cdot (z_{\theta_1}(x, c, t)x + \sqrt{\alpha_t}(1 - z_{\theta_1}(x, c, t))h_{\theta_2}(x, c, t)) . \quad (28)$$

In (27), we add a guidance strength $\lambda_{\text{model}} \geqslant 1$ on the side network as a hyper-parameter. Motivated by (25) (26), we parameterize $z_{\theta_1}$ and $h_{\theta_2}$ with a shared backbone $s_\theta$ with cross-attention blocks. In practice, we do not explicitly sample $\omega_i$ (which makes the batch size too large); instead we let the feature directions be learnable (implemented implicitly by the cross-attention projections). Importantly, our Diff-Con parameterization is orthogonal to the choice of learning algorithms: besides RLFT updates, it can also be trained with standard supervised finetuning (SFT) loss (10) when paired data are available from the target distribution.

# 5. Experiments

## 5.1. Setup

We test the proposed algorithms and parameterization under three finetuning algorithms for text-to-image generation:

1. **SFT**: supervised finetuning. The training is identical to CFG, and we conduct it to demonstrate the effectiveness of our parameterization. We use the "winner images" and their prompts from the Human Preference Dataset (HPD) v2 (Wu et al., 2023) as the training data. We train with a batch size of 128 for 1000 iterations.

2. **RWL**: finetuning with reward-weighted loss. We optimize over the Human Preference Score (HPS) v2 (Wu et al., 2023) reward model. We use the polynomial reward weighting (21) with $\tau = \tau_{\text{RWL}} =$5e-4, $\alpha = 1 + \tau_{\text{RWL}}$, and set the baseline $b$ as the mean reward of the first training batch. We train with a batch size of 64 for 2000 iterations, sampling online every 2 iterations: 64 prompts generate 64 images with the current score, then we take two gradient steps on the resulting pairs.

3. **PPO**: finetuning with KL-regularized online PPO. We optimize over the HPS-v2 reward model. We set $\tau = \tau_{\text{KL}}$ (which is the KL regularization coefficient here) to be 1e-4. We train with a batch size of 64 for 2400 iterations.

We use Stable Diffusion v1.4 (Rombach et al., 2022) as the pretrained model. In all methods, we keep the text encoder and VAE fixed, and only finetune the latent score function. For all algorithms, we set the CFG guidance strength $\lambda_{\text{CFG}}$ to be 7.5 at inference time, and the dropout probability $p_{\text{drop}}$ to be 0.1. The optimizer is Adam (Kingma & Ba, 2015).

To verify the effectiveness of our parameterization, we implement each algorithm with 5 different network structures:

- **DiffCon** (*gray-box*, ours): we use a side network $s_\theta = (z_{\theta_1}, f_{\theta_2})$ parameterized and combined with the pretrained score function as specified in (27) with $\lambda_{\text{model}} = 1.0$. $s_\theta$ consists of 2D-transformer (Vaswani et al., 2017) blocks and ResNet (He et al., 2016) blocks, and we take one (linearly transformed) output channel at the final layer to be $z_{\theta_1}$ (yielding a scalar), and use the remaining channels as $f_{\theta_2}$. The structure details are in Appendix C. The learning rate under DiffCon is 1e-5 for all algorithms. We initialize the weights of $s_\theta$'s last layer to 0 so that, same as LoRA, the initial output is the same as the pretrained model's output.

- **DiffCon-Naive** (*gray-box* baseline): we use a side network $\overline{s}_\theta$ as in DiffCon that's almost the same as $s_\theta$, except that we do not split the CNN channels at the final layer as is done in DiffCon. Instead, we directly set $\epsilon_\theta(x_t, c, t) = \epsilon_0(x_t, c, t) + \lambda_{\text{model}}\overline{s}_\theta(x_t, c, t)$.

  See Appendix C for a more detailed comparison between DiffCon and DiffCon-Naive. The learning rate under DiffCon-Naive is 1e-5 for all algorithms.

- **LoRA** (*white-box* baseline): LoRA finetuning with rank $r = 16$. We set the learning rate under LoRA to be 1e-5 for SFT and 1e-4 for RWL and PPO.

- **DiffCon-J** (*white-box*, ours): we inject LoRA layers with rank $r = 4$ into the pretrained model and also add Diff-Con and jointly train both. For SFT, we set the learning rate for both LoRA and DiffCon to be 1e-5. For RWL and PPO, we set the learning rates for LoRA and DiffCon to be 1e-4 and 1e-5 respectively.

- **DiffCon-S** (*white-box*, ours): similar to DiffCon-J but we train LoRA and DiffCon separately and combine them at evaluation time.

**Evaluation.** For automatic evaluation of the finetuned models, the main metric is the HPS-v2 win rate (against the pretrained model or the baselines). We also use CLIP (Radford et al., 2021), PickScore (Kirstain et al., 2023) and CLIP-Aesthetics (Wu et al., 2023) to demonstrate they are not compromised during our finetuning. We found at test time, enlarging the side network guidance strength $\lambda_{\text{model}}$ can help the finetuned models to achieve better performance, especially for SFT and RWL. Thus we evaluate DiffCon, DiffCon-Naive, DiffCon-J, DiffCon-S with a few $\lambda_{\text{model}}$ values, and report the best HPS-v2 win rate among them for

| Method | # Params ↓ | gray-box | HPS-v2 win rate vs. pretrained model ↑ | | |
|--------|-----------|----------|-------------------|-------------------|-------------------|
| | | | SFT (step 1000) | RWL (step 2000) | PPO (step 2400) |
| Pretrained | - | - | 0.500 | 0.500 | 0.500 |
| **DiffCon (ours)** | $1.2 \times 10^7$ | ✓ | **0.6667 ± 0.0028** | **0.6815 ± 0.0111** | **0.6957 ± 0.0152** |
| DiffCon-Naive | $1.2 \times 10^7$ | ✓ | 0.5655 ± 0.0165 | 0.5060 ± 0.0190 | 0.5201 ± 0.0203 |
| LoRA | $1.7 \times 10^7$ | ✗ | 0.5766 ± 0.0137 | 0.6109 ± 0.0059 | 0.9048 ± 0.0084 |
| **DiffCon-J (ours)** | $1.6 \times 10^7$ | ✗ | **0.6964 ± 0.0293** | **0.6555 ± 0.0210** | **0.9353 ± 0.0079** |
| **DiffCon-S (ours)** | $1.6 \times 10^7$ | ✗ | **0.6964 ± 0.0018** | **0.7091 ± 0.0155** | **0.9315 ± 0.0164** |

*Table 1.* HPS-v2 win rate against the pretrained model on the HPS-v2 test prompt set for all setups (each at its reported checkpoint).

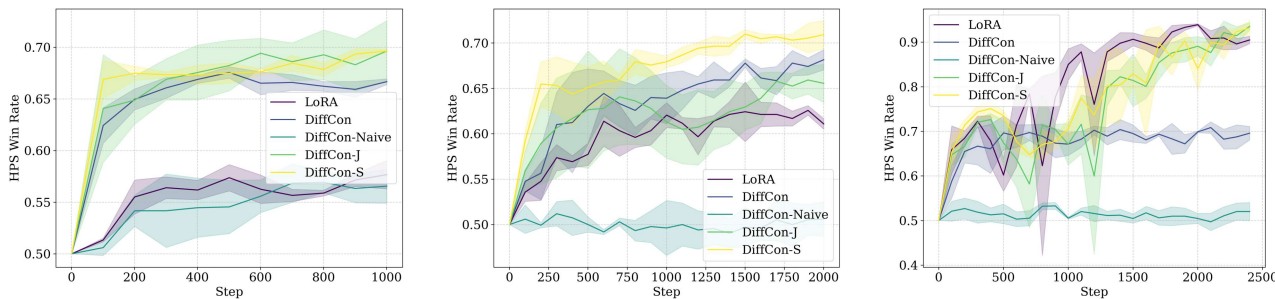

*Figure 2.* Curves of HPS-v2 win rate against the pretrained model for SFT (left), RWL (middle), and PPO (right).

each checkpoint. We also conduct human evaluation on PPO. See Appendix C for more details.

## 5.2. Results

**Main results.** Table 1 reports the HPS-v2 win rate improvements against the pretrained model at the end of training for each algorithm, Figure 2 shows the HPS-v2 win rate curves for each algorithm, and Figure 3 shows the win rate comparisons for both our gray-box and white-box methods against their own baselines. They all demonstrate that both our gray-box and white-box methods can surpass their own baselines. Notably, for SFT and RWL, our gray-box method DiffCon (with fewer parameters) can surpass the white-box LoRA, which requires access to the model's internals. For PPO, our white-box methods can reach a very high win rate ($> 0.9$) against the pretrained model. Some generation examples can be found in Table 5, 6, 7 in Appendix D.

**More results and ablation studies.** Additional results are in Appendix D. Figures 4, 5, 6, and 7 show that increasing the test-time guidance strength $\lambda_{\text{model}}$ improves generation quality. CLIP, CLIP-Aesthetics, and PickScore sanity checks for the pretrained model and all methods are also provided (Tables 8, 9, 10). We also compare DPOK's reward-weight loss (Fan et al., 2023, Algorithm 2) with our polynomial reward weighting (21) (Figure 8), and ablate key hyperparameters (e.g., $\tau_{\text{KL}}$, $\tau_{\text{RWL}}$, reweighting choice (20), learning rates, and side-network architecture).

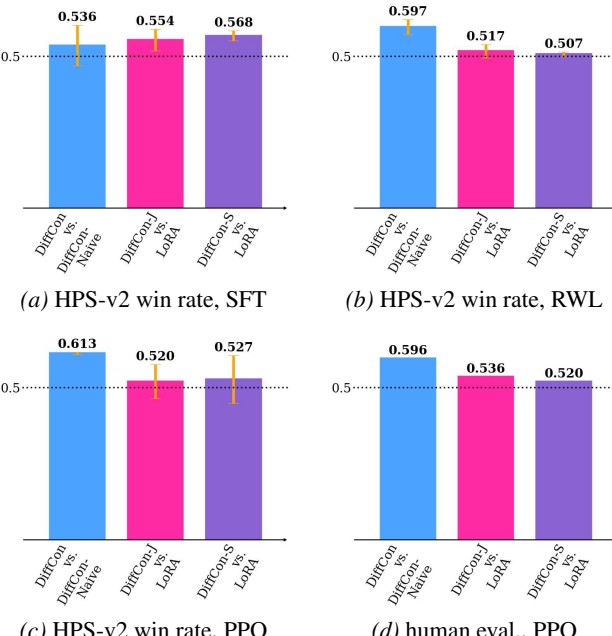

*Figure 3.* End-of-training win-rate vs. baselines. Each subplot reports three paired comparisons: (gray-box) DiffCon vs. DiffCon-Naive; (white-box) DiffCon-J vs. LoRA; (white-box) DiffCon-S vs. LoRA. (a)-(c): HPS-v2 win rates for SFT/RWL/PPO with orange error bars showing standard deviation; (d): human-evaluated win rate for PPO.

# 6. Conclusion

We presented DiffCon, a control-theoretic view of reverse diffusion that casts sampling as state-only stochastic control in a (generalized) linearly-solvable MDP. This perspective formalizes control as reweighting the pretrained reverse-time dynamics under an $f$-divergence regularizer, yielding practical fine-tuning objectives that unify supervised and reward-driven adaptation, including PPO-style updates and reward-weighted regression. DiffCon also motivates a simple pretrained plus controller parameterization: a lightweight side network conditioned on intermediate denoising signals, enabling effective gray-box fine-tuning with the backbone frozen. This improves text-to-image alignment quality–efficiency trade-offs on Stable Diffusion v1.4.

A promising direction is to extend the experiments of DiffCon beyond text-to-image alignment to broader diffusion control settings such as personalization, safety alignment, and transfer learning.

## Acknowledgment

The work of Y. Chi is supported in part by NSF grant ECCS-2537078. The work of T. Yang is supported in part by the Wei Shen and Xuehong Zhang Presidential Fellowship at Carnegie Mellon University. The authors thank Krishnamurthy Dvijotham for valuable discussions.

## Impact Statement

This work develops methods for controllable fine-tuning of diffusion models. The proposed framework may improve the efficiency and reliability of adapting generative models to user-specified objectives, which can support beneficial applications such as creative tools, personalization, and safer model alignment. As with other generative modeling techniques, these methods could also be misused to produce misleading, harmful, or copyrighted content; deployment should therefore be accompanied by appropriate safeguards, dataset governance, and evaluation for unintended behaviors.

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

# A. Additional Details on RLFT Algorithms

**Relationship to existing RWL methods for diffusion finetuning.** Lee et al. (2023) proposes a heuristic reward-weighted loss for finetuning the diffusion model: they minimize a reward-weighted negative log-likelihood (NLL) on the generated data weighted by reward $r(x_T, c)$ itself. Fan et al. (2023, Algorithm 2) also uses a loss that is an upper bound of the reward-weighted NLL with the same linear weighting.

Uehara et al. (2024a, Section 6.1) proposes a reward-weighted MLE algorithm with the following loss (under our notation):

$$
\mathbb{E}_{\substack{c,t,\xi \\ x_{1:t-1} \sim p_\theta, \\ x_{t:T} \sim p_0}} \left[ \exp\left( \frac{r(x_T, c)}{\tau} \right) \left\| \sqrt{\frac{\beta_t}{\alpha_t(1 - \overline{\alpha}_t)}} (\epsilon_\theta(x_t, c, t) - \epsilon_0(x_t, c, t)) + \xi \right\|_2^2 \right], \tag{29}
$$

where $c \sim p_c$, $t \sim \mathcal{U}([T-1])$, $\xi \sim \mathcal{N}(0, \mathbf{I}_d)$, $\epsilon_\theta$ is their (unconditional) score function, and we use $p_\theta$ to denote the transition kernel under the current parameter $\theta$. We can see that their score function could be seen as a (linear) reparametrization of our (unconditional) score function $\epsilon_\theta^\star$. And they use a mixture of online and offline sampling.

**Relationship to existing policy gradient-type methods for diffusion finetuning.** Prior RL fine-tuning works such as DDPO (Black et al., 2023) and DPOK (Fan et al., 2023) propose policy gradient methods under the MDP framework described at the end of Section 3.1, which takes the policy to be the reverse transition kernel $p_{\theta,t}(x_{t+1} \mid x_t, c)$. Converting to our notation, the following relationships hold:

1. When $\tau = 0$, our policy-gradient reduces to the standard REINFORCE gradient used in DDPO:

$$
\nabla_\theta \mathcal{J}_\theta^{\text{DDPO}} = \mathbb{E}_{\substack{c \sim p_c, \\ x_{1:T} \sim p_\theta(\cdot \mid c)}} \left[ \sum_{t=0}^{T-1} \nabla_\theta \log p_{\theta,t}(x_{t+1} \mid x_t, c) \, r(x_T, c) \right] \tag{30}
$$

   up to the usual addition of baselines (which do not change the expectation of the gradient).

2. DPOK (Fan et al., 2023) also incorporates a regularization term toward the pretrained model. However, DPOK optimizes a different surrogate objective (based on tractable approximations/upper bounds for the KL-to-reference term), which leads to a policy gradient with a different regularization term than ours under KL divergence:

$$
\nabla_\theta \mathcal{J}_\theta^{\text{DPOK}} = \mathbb{E}_{\substack{c \sim p_c, \\ x_{1:T} \sim p_\theta(\cdot \mid c)}} \left[ \sum_{t=0}^{T-1} \nabla_\theta \log p_{\theta,t}(x_{t+1} \mid x_t, c) \left( r(x_T, c) - \tau \log \frac{p_{\theta,t}(x_{t+1} \mid x_t, c)}{p_{0,t}(x_{t+1} \mid x_t, c)} \right) \right]. \tag{31}
$$

   In comparison, by (15) we can see that when $D_f = D_{\text{KL}}$, our regularization term at each step $t$ is a sum

$$
\tau \sum_{s=t}^{T-1} \log \frac{p_{\theta,s}(x_{s+1} \mid x_s, c)}{p_{0,s}(x_{s+1} \mid x_s, c)}. \tag{32}
$$

# B. Missing Proofs

**Notation.** We usually use $q$ to denote the distributions induced by the forward process, and use $p$ to denote the distributions induced by the reverse process.

## B.1. Proof of Proposition 1

This proof follows a similar idea to the proof of Lemma 6 in Cen et al. (2022). We let

$$
D_{\theta,t} \coloneqq D_f\left(p_{\theta,t}(\cdot \mid x_t, c) \,\|\, p_{0,t}(\cdot \mid x_t, c)\right)
$$

for notation simplicity. By (14), we have for any $c \in \mathcal{C}$:

$$
\begin{aligned}
\nabla_\theta V_{\theta,0}(x_0, c) &= \nabla_\theta \left[ \int_{x_1} p_{\theta,0}(x_1|x_0, c) \left( r_0(x_0, c) + V_{\theta,1}(x_1, c) \right) dx_1 \right] - \tau \nabla_\theta D_{\theta,0} \\
&= \int_{x_1} \left( p_{\theta,0}(x_1|x_0, c) \nabla_\theta \log p_{\theta,0}(x_1|x_0, c) \right) \left( r_0(x_0, c) + V_{\theta,1}(x_1, c) \right) dx_1 - \tau \nabla_\theta D_{\theta,0} \\
&\quad + \int_{x_1} p_{\theta,0}(x_1|x_0, c) \nabla_\theta V_{\theta,1}(x_1, c) dx_1.
\end{aligned}
\tag{33}
$$

Note that for all $t = 0, 1, \cdots, T-1$, we have

$$
\int_{x_{t+1}} p_{\theta,t}(x_{t+1}|x_t, c) \nabla_\theta \log p_{\theta,t}(x_{t+1}|x_t, c) \, dx_{t+1} = \nabla_\theta \int_{x_{t+1}} p_{\theta,t}(x_{t+1}|x_t, c) \, dx_{t+1} = \nabla_\theta 1 = 0,
\tag{34}
$$

and (recall we define $\zeta_{\theta,s+1} := \frac{p_{\theta,s}(x_{s+1}|x_s, c)}{p_{0,s}(x_{s+1}|x_s, c)}$)

$$
\begin{aligned}
\nabla_\theta D_{\theta,t} &= \int f'(\zeta_{\theta,t+1}) \nabla_\theta p_{\theta,t}(x_{t+1}|x_t, c) \, dx_{t+1} \\
&= \int p_{\theta,t}(x_{t+1}|x_t, c) f'(\zeta_{\theta,t+1}) \nabla_\theta \log p_{\theta,t}(x_{t+1}|x_t, c) \, dx_{t+1}.
\end{aligned}
\tag{35}
$$

Letting $t = 0$ in (34), (35), and plugging them into (33), we have

$$
\nabla_\theta V_{\theta,0}(x_0, c) = \mathbb{E}_{x_1 \sim p_{\theta,0}(\cdot|x_0, c)} \left[ \nabla_\theta \log p_{\theta,0}(x_1|x_0, c) \left( -\tau f'(\zeta_{\theta,1}) + V_{\theta,1}(x_1, c) \right) + \nabla_\theta V_{\theta,1}(x_1, c) \right],
\tag{36}
$$

where $p_{\theta,0}(x_1|x_0, c) = p_{\theta,0}(x_1|c)$. Repeating the above process for $t = 1, \ldots, T-1$, and taking expectation over $c \sim p_c$, we obtain the following policy gradient:

$$
\begin{aligned}
\nabla_\theta J_\theta &= \nabla_\theta \mathbb{E}_{c \sim p_c} \left[ V_{\theta,1}(x_1, c) \right] \\
&= \mathbb{E}_{\substack{c \sim p_c, \\ x_{1:T}|p_\theta, c}} \left[ \sum_{t=0}^{T-1} \left( \nabla_\theta \log p_{\theta,t}(x_{t+1}|x_t, c) \right) \underbrace{\left( V_{\theta,t+1}(x_{t+1}, c) - \tau f'(\zeta_{\theta,t+1}) \right)}_{Q_{\theta,t+1}} \right],
\end{aligned}
\tag{37}
$$

We define the advantage function as

$$
\begin{aligned}
A_{\theta,t+1} &:= Q_{\theta,t+1} - \mathbb{E}_{x_{t+1} \sim p_{\theta,t}(\cdot|x_t, c)} \left[ Q_{\theta,t+1} \right] \\
&= V_{\theta,t+1}(x_{t+1}, c) - \tau f'(\zeta_{\theta,t+1}) - \left( \mathbb{E}_{x_{t+1} \sim p_{\theta,t}(\cdot|x_t, c)} \left[ V_{\theta,t+1}(x_{t+1}, c) \right] - \tau \mathbb{E}_{x_{t+1} \sim p_{\theta,t}(\cdot|x_t, c)} \left[ f'(\zeta_{\theta,t+1}) \right] \right) \\
&\stackrel{(14)}{=} V_{\theta,t+1}(x_{t+1}, c) - \tau f'(\zeta_{\theta,t+1}) - V_{\theta,t}(x_t, c) - \tau D_f \left( p_{\theta,t}(\cdot|x_t, c) \,\|\, p_{0,t}(\cdot|x_t, c) \right) \\
&\quad + \tau \mathbb{E}_{x_{t+1} \sim p_{\theta,t}(\cdot|x_t, c)} \left[ f'(\zeta_{\theta,t+1}) \right],
\end{aligned}
$$

which gives (16).

$$
\nabla_\theta J_\theta = \mathbb{E}_{\substack{c \sim p_c, \\ x_{1:T}|p_\theta, c}} \left[ \sum_{t=0}^{T-1} \left( \nabla_\theta \log p_{\theta,t}(x_{t+1}|x_t, c) \right) A_{\theta,t+1} \right]
\tag{38}
$$

is guaranteed by (34).

### B.2. Proof of Theorem 1

**Step 1: Prove the equivalence between the reward-weighted loss (19) and $\mathcal{L}_{\mathrm{org}}(\cdot; \widetilde{p}^\star)$ under $f$-divergence.** We'll utilize the following lemma in this step, which is adapted from Proposition 3.1 in Ma et al. (2025):

**Lemma 1** *Given any target distribution $p^\star(\cdot|c)$ of $x_T$, let $q_t^\star(\cdot|c)$ be the forward marginal distribution of $x_t := \sqrt{\bar{\alpha}_t} x_T + \sqrt{1 - \bar{\alpha}_t}\xi$ with $\xi \sim \mathcal{N}(0, \mathbf{I}_d)$, $x_T \sim p^\star(\cdot|c)$. Then for any $g : \mathbb{R}^d \times \mathcal{C} \times [T] \to (0, +\infty)$, $\epsilon$ and $c \in \mathcal{C}$, we define*

$$\mathcal{L}(\epsilon) := \frac{1}{2} \mathbb{E}_{c \sim p_c, t \sim \mathcal{U}([T-1])} \left[ \int g(x_t, c, t) \left\| \epsilon(x_t, c, t) + \sqrt{1 - \bar{\alpha}_t} \nabla_{x_t} \log q_t^\star(x_t|c) \right\|_2^2 dx_t \right]. \tag{39}$$

*Then for any $g : \mathbb{R}^d \times \mathcal{C} \times [T] \to (0, +\infty)$ that makes (39) well-defined, $\mathcal{L}_{\text{org}}(\cdot; p^\star)$ and $\mathcal{L}$ has the same minimum $\epsilon_{p^\star}$, which satisfies*

$$\forall (x_t, c, t) \in \mathbb{R}^d \times \mathcal{C} \times [T]: \quad \epsilon_{p^\star}(x_t, c, t) = -\sqrt{1 - \bar{\alpha}_t} \nabla_{x_t} \log q_t^\star(x_t|c). \tag{40}$$

The proof of Lemma 1 is given in Appendix B.4.

We define the density ratio $w$ as

$$w(x_{1:T}, c) := \frac{P(x_{1:T}|c)}{P_0(x_{1:T}|c)}. \tag{41}$$

Then we can rewrite the optimization problem (18) as

$$\max_{w(\cdot, c)} \mathbb{E}_{x_{1:T} \sim P_0(\cdot|c)} \left[ r(x_T, c) w(x_{1:T}, c) - \tau f(w(x_{1:T}, c)) \right]$$

$$\text{s.t.} \quad w(x_{1:T}, c) \geqslant 0, \quad \mathbb{E}_{x_{1:T} \sim P_0(\cdot|c)} [w(x_{1:T}, c)] = 1. \tag{42}$$

We let $w^\star$ be the optimal solution of (42).

Introduce a multiplier $b(c) \in \mathbb{R}$ for the constraint $\mathbb{E}_{x_{1:T} \sim P_0(\cdot|c)}[w(x_{1:T}, c)] = 1$. The Lagrangian is

$$L(w(x_{1:T}, c), b(c)) = \mathbb{E}_{x_{1:T} \sim P_0(\cdot|c)} \left[ r(x_T, c) w(x_{1:T}, c) - \tau f(w(x_{1:T}, c)) - b(c) w(x_{1:T}, c) \right] + b(c). \tag{43}$$

We consider Gateaux derivative and take $w \mapsto w + \epsilon h$ with arbitrary bounded $h$. Differentiating at $\epsilon = 0$, we have

$$\left. \frac{d}{d\epsilon} L(w + \epsilon h, b) \right|_{\epsilon=0} = \int \underbrace{\{r(x_T, c) - \tau f'(w(x_{1:T}, c)) - b(c)\}}_{=:\phi(x_{1:T}, c)} h(x_{1:T}) P_0(x_{1:T}|c) dx_{1:T}. \tag{44}$$

By KKT condition, if $w^\star(x_{1:T}, c) > 0$, $\phi(x_{1:T}, c) = 0$; if $w^\star(x_{1:T}, c) = 0$, $\phi(x_{1:T}, c) \leqslant 0$ (no profitable increase). Therefore, we have

$$w^\star(x_{1:T}, c) = \left[ (f')^{-1} \left( \frac{r(x_T, c) - b_{f,\tau}(c)}{\tau} \right) \right]_+, \tag{45}$$

for some $b_{f,\tau}(c) \in \mathbb{R}$, where $[x]_+ := \max(0, x)$, and $b_{f,\tau}(c)$ is chosen so that

$$\mathbb{E}_{x_{1:T} \sim P_0(\cdot|c)} \left[ (f')^{-1} \left( \frac{r(x_T, c) - b_{f,\tau}(c)}{\tau} \right) \right]_+ = 1.$$

This suggests $\widetilde{p}^\star(x_T|c)$ satisfies

$$\widetilde{p}^\star(x_T|c) = \int \widetilde{P}^\star(x_{1:T}|c) dx_{1:T-1}$$

$$= \int P_0(x_{1:T}|c) dx_{1:T-1} \left[ (f')^{-1} \left( \frac{r(x_T, c) - b_{f,\tau}(c)}{\tau} \right) \right]_+$$

$$= p_{0,T}(x_T|c) \left[ (f')^{-1} \left( \frac{r(x_T, c) - b_{f,\tau}(c)}{\tau} \right) \right]_+, \tag{46}$$

where $p_{0,T}(x_T|c)$ is the reverse marginal distribution of $x_T$ under the pretrained model. Let

$$g(x_t, c, t) = \widetilde{q}_t^\star(x_t|c) \tag{47}$$

in Lemma 1, where $\widetilde{q}_t^\star (x_t|c)$ is the forward marginal distribution of $x_t = \sqrt{\overline{\alpha}_t} x_T + \sqrt{1-\overline{\alpha}_t}\xi$ with $\xi \sim \mathcal{N}(0, \mathbf{I}_d)$ and $x_T \sim \widetilde{p}^\star(\cdot|c)$. Then we have

$$\int g(x_t, c, t) \left\| \epsilon(x_t, c, t) + \sqrt{1-\overline{\alpha}_t} \nabla_{x_t} \log \widetilde{q}_t^\star (x_t|c) \right\|_2^2 dx_t$$

$$= \int \frac{g(x_t, c, t)}{\widetilde{q}_t^\star(x_t|c)} \int \widetilde{p}^\star(x_T|c) q_t(x_t|x_T) \left\| \epsilon(x_t, c, t) + \sqrt{1-\overline{\alpha}_t} \nabla_{x_t} \log q_t(x_t|x_T) \right\|_2^2 dx_T dx_t + \mathsf{const}$$

$$= \int \int p_{0,T}(x_T|c) \left[ (f')^{-1} \left( \frac{r(x_T, c) - b_{f,\tau}(c)}{\tau} \right) \right]_+ q_t(x_t|x_T) \left\| \epsilon(x_t, c, t) + \sqrt{1-\overline{\alpha}_t} \nabla_{x_t} \log q_t(x_t|x_T) \right\|_2^2 dx_T dx_t + \mathsf{const}$$

$$= \mathbb{E}_{\substack{x_T \sim p_{0,T}, \\ \xi \sim \mathcal{N}(0, \mathbf{I}_d)}} \left[ \left[ (f')^{-1} \left( \frac{r(x_T, c) - b_{f,\tau}(c)}{\tau} \right) \right]_+ \left\| \xi - \epsilon \left( \sqrt{\overline{\alpha}_t} x_T + \sqrt{1-\overline{\alpha}_t}\xi, c, t \right) \right\|_2^2 \right] + \mathsf{const}, \tag{48}$$

where the second line is obtained by replacing $q^\star(x_t|c)$ and $p^\star(x_T|c)$ in (108) in the proof of Lemma 1 by $\widetilde{q}_t^\star (x_t|c)$ and $\widetilde{p}^\star (x_T|c)$, the third line is by our choice of $g$ in (47) and (46). Therefore, by Lemma 1, under (47), $\mathcal{L}(\epsilon)$ has the same minimum as $\mathcal{L}_{\mathrm{org}}(\cdot; \widetilde{p}^\star)$.

**Step 2: show $\widetilde{p}^\star(x_T|c) = p_{u^\star}(x_T|c)$ when $D_f = D_{\mathrm{KL}}$.** When $D_f = D_{\mathrm{KL}}$, the optimal KL-regularized Bellman equation is ($t = 0, \cdots, T-1$)

$$\forall c \in \mathcal{C}: \quad V_t^\star(x_t, c) = \max_{u(\cdot, x_t, c)} r_t(x_t, c) + \mathbb{E}_{x_{t+1} \sim \mathbb{P}_{u,t}(\cdot|x_t, c)} \left[ V_{t+1}^\star(x_{t+1}, c) - \tau u_t(x_{t+1}, x_t, c) \right]$$

$$= \max_{u(\cdot, x_t, c)} V_{u,t}(x_t, c), \tag{49}$$

which induces

$$\forall c \in \mathcal{C}: \quad u_t^\star(x_{t+1}, x_t, c) := \arg \max_{u(\cdot, x_t, c)} V_{u,t}(x_t, c)$$

$$= \frac{1}{\tau} V_{t+1}^\star(x_{t+1}, c) - \log \left( \mathbb{E}_{x_{t+1} \sim p_{0,t}(\cdot|x_t, c)} \left[ \exp \left( \frac{1}{\tau} V_{t+1}^\star(x_{t+1}, c) \right) \right] \right). \tag{50}$$

Then by (8), the optimal transition operator becomes

$$\mathbb{P}_{u^\star, t}(x_{t+1}|x_t, c) = p_{0,t}(x_{t+1}|x_t, c) \frac{\exp \left( \frac{1}{\tau} V_{t+1}^\star(x_{t+1}, c) \right)}{\mathbb{E}_{x_{t+1} \sim p_{0,t}(\cdot|x_t, c)} \left[ \exp \left( \frac{1}{\tau} V_{t+1}^\star(x_{t+1}, c) \right) \right]}. \tag{51}$$

In addition, plugging (50) into (49), we have

$$V_t^\star(x_t, c) = r_t(x_t, c) + \tau \log \left( \mathbb{E}_{x_{t+1} \sim p_{0,t}(\cdot|x_t, c)} \left[ \exp \left( \frac{1}{\tau} V_{t+1}^\star(x_{t+1}, c) \right) \right] \right). \tag{52}$$

Define

$$Z_t(x_t, c) := \exp \left( \frac{1}{\tau} V_t^\star(x_t, c) \right), \tag{53}$$

then under our terminal reward setting (11), we have

$$Z_T(x_T, c) := \exp \left( \frac{1}{\tau} r(x_T, c) \right), \tag{54}$$

and the following recursion given by (52):

$$\forall t = 0, \cdots, T-1: \quad Z_t(x_t, c) = \mathbb{E}_{x_{t+1} \sim p_{0,t}(\cdot|x_t, c)} \left[ Z_{t+1}(x_{t+1}, c) \right]. \tag{55}$$

Especially, when $t = 0$, we have (recall we define $x_0 := \emptyset$)

$$Z_0(c) := Z_0(x_0, c) = \mathbb{E}_{x_1 \sim \mathcal{N}(0, \mathbf{I}_d)} \left[ Z_1(x_1, c) \right]. \tag{56}$$

Plugging (55) into (51), we have for all $t = 0, \cdots, T-1$:

$$\mathbb{P}_{u^\star,t}(x_{t+1}|x_t, c) = p_{0,t}(x_{t+1}|x_t, c) \frac{Z_{t+1}(x_{t+1}, c)}{Z_t(x_t, c)}. \tag{57}$$

Let $\mathbb{P}_{u^\star}(x_{1:T}|c)$ be the joint distribution of $x_{1:T}$ under the optimal control $u^\star$. Then by (57), we have

$$
\begin{aligned}
\mathbb{P}_{u^\star}(x_{1:T}|c) &= p_{u^\star,-1}(x_0|c) \prod_{t=0}^{T-1} \mathbb{P}_{u^\star,t}(x_{t+1}|x_t, c) \\
&= \prod_{t=0}^{T-1} p_{0,t}(x_{t+1}|x_t, c) \frac{Z_{t+1}(x_{t+1}, c)}{Z_t(x_t, c)} \\
&= P_0(x_{1:T}|c) \frac{Z_T(x_T, c)}{Z_0(c)} \\
&\stackrel{(54)}{=} P_0(x_{1:T}|c) \frac{\exp\left(\frac{1}{\tau} r(x_T, c)\right)}{Z_0(c)}.
\end{aligned} \tag{58}
$$

Integrating out $x_{1:T-1}$, we have

$$p_{u^\star}(x_T|c) = p_{0,T}(x_T|c) \frac{\exp\left(\frac{1}{\tau} r(x_T, c)\right)}{Z_0(c)}, \tag{59}$$

where $p_{0,T}(x_T|c)$ is the reverse marginal distribution of $x_T$ under the pretrained model. Note KL divergence is $f$-divergence with $f(t) = t \log t$. By (46), we have

$$\widetilde{p}^\star(x_T|c) \propto p_{0,T}(x_T|c) \exp\left(\frac{1}{\tau} r(x_T, c)\right), \tag{60}$$

where we use the fact that $(f')^{-1}(y) = \exp(y-1)$ for KL divergence. This gives the desired result $\widetilde{p}^\star(x_T|c) = p_{u^\star}(x_T|c)$.

### B.3. Proof of Proposition 2

Let $q_{0,t}(x_t|c), q_{u^\star,t}(x_t|c)$ denote the marginal distributions of $x_t$ induced by the forward process starting from target distributions $p_0$ and $p_{u^\star}$, respectively, i.e., $q_{0,t}(\cdot|c)$ (resp. $q_{u^\star,t}(\cdot|c)$) is the probability of $x_t = \sqrt{\overline{\alpha}_t} x_T + \sqrt{1 - \overline{\alpha}_t} \xi_t$, where $\xi_t \sim \mathcal{N}(0, \mathbf{I}_d)$, $x_T \sim p_0(\cdot|c)$ (resp. $p_{u^\star}(\cdot|c)$). Then when $D_f = D_{\mathrm{KL}}$, we have

$$
\begin{aligned}
q_{u^\star,t}(x_t|c) &= \int q_t(x_t|x_T) p_{u^\star}(x_T|c) dx_T \\
&\stackrel{(59)}{=} \frac{1}{Z_0(c)} \int q_t(x_t|x_T) p_0(x_T|c) \exp\left(\frac{1}{\tau} r(x_T, c)\right) dx_T \\
&= \frac{1}{Z_0(c)} \int q_{0,t}(x_T|x_t, c) q_{0,t}(x_t|c) \exp\left(\frac{1}{\tau} r(x_T, c)\right) dx_T \\
&= q_{0,t}(x_t|c) \frac{\mathbb{E}_{x_T \sim q_{0,t}(\cdot|x_t, c)}\left[\exp\left(\frac{1}{\tau} r(x_T, c)\right)\right]}{Z_0(c)},
\end{aligned} \tag{61}
$$

where the third line follows from the following relation given by the Bayes' rule:

$$q_{0,t}(x_T|x_t, c) = \frac{q_t(x_t|x_T) p_0(x_T|c)}{q_{0,t}(x_t|c)}. \tag{62}$$

Thus we have

$$
\begin{aligned}
q_{u^\star,t}(x_{t+1}|x_t, c) &= \frac{q_{t+1}(x_t|x_{t+1}) q_{u^\star,t}(x_{t+1}|c)}{q_{u^\star,t}(x_t|c)} \\
&\stackrel{(61)}{=} \frac{q_{t+1}(x_t|x_{t+1}) q_{0,t+1}(x_{t+1}|c)}{q_{0,t}(x_t|c)} \frac{\mathbb{E}_{x_T \sim q_{0,t+1}(\cdot|x_{t+1}, c)}\left[\exp\left(\frac{1}{\tau} r(x_T, c)\right)\right]}{\mathbb{E}_{x_T \sim q_{0,t}(\cdot|x_t, c)}\left[\exp\left(\frac{1}{\tau} r(x_T, c)\right)\right]} \\
&= q_{0,t}(x_{t+1}|x_t, c) \frac{\mathbb{E}_{x_T \sim q_{0,t+1}(\cdot|x_{t+1}, c)}\left[\exp\left(\frac{1}{\tau} r(x_T, c)\right)\right]}{\mathbb{E}_{x_T \sim q_{0,t}(\cdot|x_t, c)}\left[\exp\left(\frac{1}{\tau} r(x_T, c)\right)\right]},
\end{aligned} \tag{63}
$$

where $q_{0,t}(x_{t+1}|x_t, c)$ and $q_{u^\star,t}(x_{t+1}|x_t, c)$ represent the posterior conditional distributions induced by the forward process starting from target distributions $p_0$ and $p_{u^\star}$, respectively.

We define

$$\psi_t(x_t, c) := \mathbb{E}_{x_T \sim q_{0,t}(\cdot|x_t, c)} \left[\exp\left(\frac{1}{\tau} r(x_T, c)\right)\right], \tag{64}$$

then by tower property of expectation, we have

$$\psi_t(x_t, c) = \mathbb{E}_{x_{t+1} \sim q_{0,t}(\cdot|x_t, c)} \left[\psi_{t+1}(x_{t+1}, c)\right], \tag{65}$$

and by (63), we have

$$q_{u^\star,t}(x_{t+1}|x_t, c) = q_{0,t}(x_{t+1}|x_t, c) \frac{\psi_{t+1}(x_{t+1}, c)}{\psi_t(x_t, c)}. \tag{66}$$

Thus we can express the mean of $q_{u^\star,t}(x_{t+1}|x_t, c)$ as follows:

$$\mathbb{E}_{x_{t+1} \sim q_{u^\star,t}(\cdot|x_t, c)} [x_{t+1}] = \frac{\mathbb{E}_{x_{t+1} \sim q_{0,t}(\cdot|x_t, c)} [x_{t+1} \psi_{t+1}(x_{t+1}, c)]}{\psi_t(x_t, c)}. \tag{67}$$

We let $p_{u^\star,t}(x_{t+1}|x_t, c)$ denote the reverse conditional distribution of $x_{t+1}$ given $x_t$ under the optimal score function $\epsilon^\star$. Then we have the following lemma that says the mean of the posterior conditional distribution $q_{u^\star,t}(\cdot|x_t, c)$ induced by the forward process is the same as the mean of the learned reverse kernel $p_{u^\star,t}(\cdot|x_t, c)$:

**Lemma 2** *For any data distribution $p(\cdot|c)$ of $x_T$ ($\forall c \in \mathcal{C}$), let $\epsilon_p^\star$ be the optimal score function that minimizes the diffusion loss $\mathcal{L}_{\mathrm{org}}(\cdot; p)$ defined in (10). Then for any $x_t \in \mathbb{R}^d, c \in \mathcal{C}, t \in [T-1]$, we have*

$$\mathbb{E}_{x_{t+1} \sim q_t(\cdot|x_t, c)} [x_{t+1}] = \mu_p^\star(x_t, c, t) := \frac{1}{\sqrt{\alpha_t}} x_t - \frac{\beta_t}{\sqrt{\alpha_t (1 - \overline{\alpha}_t)}} \epsilon_p^\star(x_t, c, t), \tag{68}$$

*where $q_t(\cdot|x_t, c)$ is the posterior conditional distribution induced by the forward process starting from data distribution $p$.*

The proof of Lemma 2 is given in Appendix B.5.

By Lemma 2, we have

$$\begin{aligned}
\mu^\star(x_t, c, t) &= \mathbb{E}_{x_{t+1} \sim q_{u^\star,t}(\cdot|x_t, c)} [x_{t+1}] \\
&\stackrel{(67)}{=} \frac{\mathbb{E}_{x_{t+1} \sim q_{0,t}(\cdot|x_t, c)} [x_{t+1} \psi_{t+1}(x_{t+1}, c)]}{\psi_t(x_t, c)} \\
&= \underbrace{\frac{\mathbb{E}_{x_{t+1} \sim p_{0,t}(\cdot|x_t, c)} [x_{t+1} \psi_{t+1}(x_{t+1}, c)]}{\mathbb{E}_{x_{t+1} \sim p_{0,t}(\cdot|x_t, c)} [\psi_{t+1}(x_{t+1}, c)]}}_{:= \widetilde{\mu}^\star(x_t, c, t)} \\
&\quad + \underbrace{\frac{\mathbb{E}_{x_{t+1} \sim q_{0,t}(\cdot|x_t, c)} [x_{t+1} \psi_{t+1}(x_{t+1}, c)]}{\psi_t(x_t, c)} - \frac{\mathbb{E}_{x_{t+1} \sim p_{0,t}(\cdot|x_t, c)} [x_{t+1} \psi_{t+1}(x_{t+1}, c)]}{\mathbb{E}_{x_{t+1} \sim p_{0,t}(\cdot|x_t, c)} [\psi_{t+1}(x_{t+1}, c)]}}_{:= \delta_0(x_t, c, t)}.
\end{aligned} \tag{69}$$

Under this decomposition and the relation between $\mu$ and $\epsilon$, we have

$$\begin{aligned}
\epsilon^\star(x_t, c, t) - \epsilon_0(x_t, c, t) &= \frac{\sqrt{\alpha_t (1 - \overline{\alpha}_t)}}{\beta_t} (\mu_0(x_t, c, t) - \mu^\star(x_t, c, t)) \\
&= \frac{\sqrt{\alpha_t (1 - \overline{\alpha}_t)}}{\beta_t} (\mu_0(x_t, c, t) - \widetilde{\mu}^\star(x_t, c, t) - \delta_0(x_t, c, t)).
\end{aligned} \tag{70}$$

Define

$$\widetilde{\psi}_t(x_t, c) := \mathbb{E}_{x_{t+1} \sim p_{0,t}(\cdot|x_t, c)} [\psi_{t+1}(x_{t+1}, c)], \tag{71}$$

then combining (69) and (70), we have

$$
\begin{aligned}
&\epsilon^{\star}(x_t, c, t) - \epsilon_0(x_t, c, t) \\
&= \frac{\sqrt{\alpha_t (1 - \overline{\alpha}_t)}}{\beta_t} \left( -\frac{\mathbb{E}_{x_{t+1} \sim p_{0,t}(\cdot | x_t, c)} \left[ x_{t+1}(\psi_{t+1}(x_{t+1}, c) - 1) \right]}{\widetilde{\psi}_t(x_t, c)} + \left( 1 - \frac{1}{\widetilde{\psi}_t(x_t, c)} \right) \mu_0(x_t, c, t) - \delta_0(x_t, c, t) \right),
\end{aligned}
$$

(72)

where we use the fact that

$$
\mu_0(x_t, c, t) = \mathbb{E}_{x_{t+1} \sim p_{0,t}(\cdot | x_t, c)} \left[ x_{t+1} \right].
$$

(73)

Recall in (23) we define the basis functions $\phi_\omega(x_t, c, t)$ as

$$
\phi_\omega(x_t, c, t) = \left( \cos \left( \frac{\omega^\top \mu_0(x_t, c, t)}{\sqrt{\widetilde{\beta}_t}} \right), \sin \left( \frac{\omega^\top \mu_0(x_t, c, t)}{\sqrt{\widetilde{\beta}_t}} \right) \right)^\top,
$$

(74)

here we also define

$$
\rho_\omega(x_{t+1}, t) = \frac{1}{\sqrt{(2\pi\widetilde{\beta}_t)^d}} \left( \cos \left( \frac{\omega^\top x_{t+1}}{\sqrt{\widetilde{\beta}_t}} \right), \sin \left( \frac{\omega^\top x_{t+1}}{\sqrt{\widetilde{\beta}_t}} \right) \right)^\top.
$$

(75)

Then we can decompose the Gaussian reverse kernel $p_{0,t}(x_{t+1} | x_t, c)$ as

$$
p_{0,t}(x_{t+1} | x_t, c) = \langle \phi_\omega(x_t, t), \rho_\omega(x_{t+1}, t) \rangle_{\mathcal{N}(\omega)},
$$

(76)

where

$$
\langle f_\omega, g_\omega \rangle_{\mathcal{N}(\omega)} := \mathbb{E}_{\omega \sim \mathcal{N}(\cdot | 0, \mathbf{I}_d)} \left[ f_\omega^\top g_\omega \right]
$$

for any $f_\omega$ and $g_\omega$.

We define

$$
h(x_t, c, t) := \mathbb{E}_{x_{t+1} \sim p_{0,t}(\cdot | x_t, c)} \left[ x_{t+1}(\psi_{t+1}(x_{t+1}, c) - 1) \right],
$$

(77)

$$
z(x_t, c, t) := 1 - \frac{1}{\widetilde{\psi}_t(x_t, c)},
$$

(78)

then by (72) and (7),

$$
\begin{aligned}
&\epsilon^{\star}(x_t, c, t) \\
&= \frac{1}{\widetilde{\psi}_t(x_t, c)} \epsilon_0(x_t, c, t) - \frac{\sqrt{1 - \overline{\alpha}_t}}{\beta_t} \left( \frac{\sqrt{\alpha_t}}{\widetilde{\psi}_t(x_t, c)} h(x_t, c, t) + \left( 1 - \frac{1}{\widetilde{\psi}_t(x_t, c)} \right) x_t - \sqrt{\alpha_t} \delta_0(x_t, c, t) \right) \\
&= (1 - z(x_t, c, t)) \epsilon_0(x_t, c, t) - \frac{\sqrt{1 - \overline{\alpha}_t}}{\beta_t} \left( \sqrt{\alpha_t}(1 - z(x_t, c, t)) h(x_t, c, t) + z(x_t, c, t) x_t - \sqrt{\alpha_t} \delta_0(x_t, c, t) \right).
\end{aligned}
$$

(79)

Moreover, we can express $h(x_t, c, t)$ as

$$
h(x_t, c, t) = \mathbb{E}_{\omega \sim \mathcal{N}(0, \mathbf{I}_d)} \left[ \left( \underbrace{\int (\psi_{t+1}(x_{t+1}, c) - 1) x_{t+1} \rho_\omega(x_{t+1}, t)^\top dx_{t+1}}_{W^{\star}(\omega, c, t) \in \mathbb{R}^{d \times 2}} \right) \phi_\omega(x_t, c, t) \right],
$$

(80)

and express $\widetilde{\psi}_t(x_t, c) - 1 = \frac{z(x_t, c, t)}{1 - z(x_t, c, t)}$ as

$$
\frac{z(x_t, c, t)}{1 - z(x_t, c, t)} = \widetilde{\psi}_t(x_t, c) - 1 \overset{(71)}{=} \mathbb{E}_{\omega \sim \mathcal{N}(0, \mathbf{I}_d)} \left[ \left( \underbrace{\int (\psi_{t+1}(x_{t+1}, c) - 1) \rho_\omega(x_{t+1}, t)^\top dx_{t+1}}_{u^{\star}(\omega, c, t) \in \mathbb{R}^2} \right) \phi_\omega(x_t, c, t) \right],
$$

(81)

We can use random Fourier features to approximate them as

$$h(x_t, c, t) = \frac{1}{M} \sum_{i=1}^{M} W^\star(\omega_i, c, t) \phi_{\omega_i}(x_t, c, t) + \delta_h(x_t, c, t), \tag{82}$$

$$\frac{z(x_t, c, t)}{1 - z(x_t, c, t)} = \frac{1}{M} \sum_{i=1}^{M} u^\star(\omega_i, c, t)^\top \phi_{\omega_i}(x_t, c, t) + \delta_z(x_t, c, t), \tag{83}$$

where $\omega_i \overset{i.i.d.}{\sim} \mathcal{N}(0, \mathbf{I}_d)$, $i \in [M]$. Let

$$B_W := \text{ess} \sup_{\omega \in \mathbb{R}^d} \|W^\star(\omega, c, t)\|_F, \quad V_W^2 := \text{ess} \sup_{\omega \in \mathbb{R}^d} \|u^\star(\omega, c, t)\|_F^2,$$

$$B_u := \text{ess} \sup_{\omega \in \mathbb{R}^d} \|u^\star(\omega, c, t)\|_2, \quad V_u^2 := \text{ess} \sup_{\omega \in \mathbb{R}^d} \|u^\star(\omega, c, t)\|_2^2, \tag{84}$$

By Bernstein's inequality (Kwon & Perchet (2017, Proposition E.3)), we have with probability at least $1 - \delta$,

$$\|\delta_h\|_2 \leqslant \sqrt{\frac{2V_W^2 \log(2/\delta)}{M}} + \frac{2B_W \log(2/\delta)}{3M}, \quad \|\delta_z\|_2 \leqslant \sqrt{\frac{2V_u^2 \log(2/\delta)}{M}} + \frac{2B_u \log(2/\delta)}{3M}. \tag{85}$$

Finally, we bound $\delta_0(x_t, c, t)$ defined in (69).

Recall we assume $x_T \in \mathbb{B}_\infty^d(1)$, where we let

$$\mathbb{B}_\infty^d(1) := \left\{ x_T \in \mathbb{R}^d : \|x_T\|_\infty \leqslant 1 \right\}. \tag{86}$$

We let

$$r_{\min} := \inf_{x_T \in \mathbb{B}_\infty^d(1), c \in \mathcal{C}} r(x_T, c), \quad r_{\max} := \sup_{x_T \in \mathbb{B}_\infty^d(1), c \in \mathcal{C}} r(x_T, c). \tag{87}$$

Then $r_{\min}$ and $r_{\max}$ are both finite, and by (64) we know that

$$\psi_{\min} := \exp\left(\frac{r_{\min}}{\tau}\right) \leqslant \psi_t(x_t, c) \leqslant \psi_{\max} := \exp\left(\frac{r_{\max}}{\tau}\right). \tag{88}$$

we define weighting function

$$w_t(x_t, c) := \frac{\psi_t(x_t, c)}{\psi_{\min}}, \quad \overline{w}_t(x_t, c) := w_t(x_t, c) - 1, \tag{89}$$

Then we have

$$\begin{aligned}
&\delta_0(x_t, c, t) \\
&:= \frac{\mathbb{E}_{x_{t+1} \sim q_{0,t}(\cdot | x_t, c)} [x_{t+1} w_{t+1}(x_{t+1}, c)]}{\mathbb{E}_{x_{t+1} \sim q_{0,t}(\cdot | x_t, c)} [w_{t+1}(x_{t+1}, c)]} - \frac{\mathbb{E}_{x_{t+1} \sim p_{0,t}(\cdot | x_t, c)} [x_{t+1} w_{t+1}(x_{t+1}, c)]}{\mathbb{E}_{x_{t+1} \sim p_{0,t}(\cdot | x_t, c)} [w_{t+1}(x_{t+1}, c)]} \\
&= \frac{\mathbb{E}_{x_{t+1} \sim q_{0,t}(\cdot | x_t, c)} [(x_{t+1} - \mu_0(x_t, c, t)) w_{t+1}(x_{t+1}, c)]}{\mathbb{E}_{x_{t+1} \sim q_{0,t}(\cdot | x_t, c)} [w_{t+1}(x_{t+1}, c)]} - \frac{\mathbb{E}_{x_{t+1} \sim p_{0,t}(\cdot | x_t, c)} [(x_{t+1} - \mu_0(x_t, c, t)) w_{t+1}(x_{t+1}, c)]}{\mathbb{E}_{x_{t+1} \sim p_{0,t}(\cdot | x_t, c)} [w_{t+1}(x_{t+1}, c)]} \\
&= \frac{\mathbb{E}_{x_{t+1} \sim q_{0,t}(\cdot | x_t, c)} [(x_{t+1} - \mu_0(x_t, c, t)) \overline{w}_{t+1}(x_{t+1}, c)]}{\mathbb{E}_{x_{t+1} \sim q_{0,t}(\cdot | x_t, c)} [w_{t+1}(x_{t+1}, c)]} - \frac{\mathbb{E}_{x_{t+1} \sim p_{0,t}(\cdot | x_t, c)} [(x_{t+1} - \mu_0(x_t, c, t)) \overline{w}_{t+1}(x_{t+1}, c)]}{\mathbb{E}_{x_{t+1} \sim p_{0,t}(\cdot | x_t, c)} [w_{t+1}(x_{t+1}, c)]}
\end{aligned} \tag{90}$$

where the first line uses (69) and (65), and the third line follows from the fact that

$$\mu_0(x_t, c, t) = \mathbb{E}_{x_{t+1} \sim p_{0,t}(\cdot | x_t, c)} [x_{t+1}] = \mathbb{E}_{x_{t+1} \sim q_{0,t}(\cdot | x_t, c)} [x_{t+1}] \tag{91}$$

given by the optimality of $\epsilon_0$ and Lemma 2. Define

$$
\begin{aligned}
A_{q,t} &:= \mathbb{E}_{x_{t+1} \sim q_{0,t}(\cdot|x_t,c)} \left[ (x_{t+1} - \mu_0(x_t,c,t)) \overline{w}_{t+1}(x_{t+1},c) \right], \\
A_{p,t} &:= \mathbb{E}_{x_{t+1} \sim p_{0,t}(\cdot|x_t,c)} \left[ (x_{t+1} - \mu_0(x_t,c,t)) \overline{w}_{t+1}(x_{t+1},c) \right], \\
B_{q,t} &:= \mathbb{E}_{x_{t+1} \sim q_{0,t}(\cdot|x_t,c)} \left[ w_{t+1}(x_{t+1},c) \right], \\
B_{p,t} &:= \mathbb{E}_{x_{t+1} \sim p_{0,t}(\cdot|x_t,c)} \left[ w_{t+1}(x_{t+1},c) \right],
\end{aligned}
\tag{92}
$$

Then

$$
\begin{aligned}
\|\delta_0(x_t,c,t)\|_2 &= \left\| \frac{A_{q,t}}{B_{q,t}} - \frac{A_{p,t}}{B_{p,t}} \right\|_2 \\
&\leqslant \left\| \frac{A_{q,t} - A_{p,t}}{B_{q,t}} \right\|_2 + \|A_{p,t}\|_2 \left| \frac{1}{B_{p,t}} - \frac{1}{B_{q,t}} \right| \\
&\leqslant \underbrace{\|A_{q,t} - A_{p,t}\|_2}_{(i)} + \underbrace{\|A_{p,t}\|_2 |B_{p,t} - B_{q,t}|}_{(ii)},
\end{aligned}
\tag{93}
$$

where we use the fact that

$$
B_{p,t} \geqslant 1, \quad B_{q,t} \geqslant 1
$$

by our choice of $w_{t+1}$ (c.f. (89)). Below we bound $(i)$ and $(ii)$ in (93) separately utilizing the standard TV inequality:

$$
\left\| \mathbb{E}_p\left[g\right] - \mathbb{E}_q\left[g\right] \right\|_2 \leqslant \sqrt{2\mathrm{TV}\left(p,q\right)} \sqrt{\mathbb{E}_p\left[\|g\|_2^2\right] + \mathbb{E}_q\left[\|g\|_2^2\right]}.
\tag{94}
$$

for any vector-valued function $g$ and probability measure $p$ and $q$.

Define

$$
g_{t+1} := (x_{t+1} - \mu_0(x_t,c,t)) \overline{w}_{t+1}(x_{t+1},c),
\tag{95}
$$

then by (89) we have

$$
\|g_{t+1}\|_2^2 \leqslant \left( \frac{\psi_{\max}}{\psi_{\min}} - 1 \right)^2 \|x_{t+1} - \mu_0(x_t,c,t)\|_2^2.
\tag{96}
$$

and by (94) we have

$$
\begin{aligned}
(i) &= \left\| \mathbb{E}_{x_{t+1} \sim q_{0,t}(\cdot|x_t,c)}\left[g_{t+1}\right] - \mathbb{E}_{x_{t+1} \sim p_{0,t}(\cdot|x_t,c)}\left[g_{t+1}\right] \right\|_2 \\
&\leqslant \sqrt{2\mathrm{TV}\left(q_{0,t},p_{0,t}\right)} \sqrt{\mathbb{E}_{x_{t+1} \sim q_{0,t}(\cdot|x_t,c)}\left[\|g_{t+1}\|_2^2\right] + \mathbb{E}_{x_{t+1} \sim p_{0,t}(\cdot|x_t,c)}\left[\|g_{t+1}\|_2^2\right]} \\
&\overset{(96)}{\leqslant} \left( \frac{\psi_{\max}}{\psi_{\min}} - 1 \right) \sqrt{2\mathrm{TV}\left(q_{0,t},p_{0,t}\right)} \sqrt{\mathbb{E}_{x_{t+1} \sim q_{0,t}(\cdot|x_t,c)}\left[\|x_{t+1} - \mu_0(x_t,c,t)\|_2^2\right] + \mathbb{E}_{x_{t+1} \sim p_{0,t}(\cdot|x_t,c)}\left[\|x_{t+1} - \mu_0(x_t,c,t)\|_2^2\right]},
\end{aligned}
\tag{97}
$$

where we write $\mathrm{TV}\left(q_{0,t},p_{0,t}\right)$ as a shorthand for $\mathrm{TV}\left(q_{0,t}(\cdot|x_t,c), p_{0,t}(\cdot|x_t,c)\right)$. For (ii), by (96) we have

$$
\|A_{p,t}\|_2 \leqslant \left( \frac{\psi_{\max}}{\psi_{\min}} - 1 \right) \sqrt{\mathbb{E}_{x_{t+1} \sim p_{0,t}(\cdot|x_t,c)}\left[\|x_{t+1} - \mu_0(x_t,c,t)\|_2^2\right]},
\tag{98}
$$

and by (94) we have

$$
\begin{aligned}
|B_{p,t} - B_{q,t}| &\leqslant \sqrt{2\mathrm{TV}\left(q_{0,t},p_{0,t}\right)} \sqrt{\mathbb{E}_{x_{t+1} \sim q_{0,t}(\cdot|x_t,c)}\left[\|w_{t+1}(x_{t+1},c)\|_2^2\right] + \mathbb{E}_{x_{t+1} \sim p_{0,t}(\cdot|x_t,c)}\left[\|w_{t+1}(x_{t+1},c)\|_2^2\right]} \\
&\leqslant \sqrt{2} \frac{\psi_{\max}}{\psi_{\min}} \sqrt{2\mathrm{TV}\left(q_{0,t},p_{0,t}\right)}.
\end{aligned}
\tag{99}
$$

Combining The above two expressions, we have

$$(ii) \leqslant \sqrt{2} \frac{\psi_{\max}}{\psi_{\min}} \left( \frac{\psi_{\max}}{\psi_{\min}} - 1 \right) \sqrt{\mathbb{E}_{x_{t+1} \sim q_{0,t}(\cdot|x_t,c)} \left[ \|x_{t+1} - \mu_0(x_t,c,t)\|_2^2 \right]} \sqrt{2\mathrm{TV}\left(q_{0,t}, p_{0,t}\right)} \tag{100}$$

Combining (97) and (100), we have

$$\|\delta_0(x_t, c, t)\|_2 \leqslant \left( \frac{\psi_{\max}}{\psi_{\min}} - 1 \right) \left( 1 + \sqrt{2} \frac{\psi_{\max}}{\psi_{\min}} \right) \sqrt{2\mathrm{TV}\left(q_{0,t}, p_{0,t}\right)}$$
$$\cdot \sqrt{\underbrace{\mathbb{E}_{x_{t+1} \sim q_{0,t}(\cdot|x_t,c)} \left[ \|x_{t+1} - \mu_0(x_t,c,t)\|_2^2 \right]}_{(a)} + \underbrace{\mathbb{E}_{x_{t+1} \sim p_{0,t}(\cdot|x_t,c)} \left[ \|x_{t+1} - \mu_0(x_t,c,t)\|_2^2 \right]}_{(b)}}. \tag{101}$$

For (b), since $q_{0,t}(\cdot|x_t, c) = \mathcal{N}\left( \cdot \Big| \mu_0(x_t, c, t), \widetilde{\beta}_t \mathbf{I}_d \right)$, we have

$$(b) := \mathbb{E}_{x_{t+1} \sim p_{0,t}(\cdot|x_t,c)} \left[ \|x_{t+1} - \mu_0(x_t,c,t)\|_2^2 \right] = d\widetilde{\beta}_t, \tag{102}$$

For (a), we have

$$(a) = \mathbb{E}_{x_{t+1} \sim q_{0,t}(\cdot|x_t,c)} \left[ \|x_{t+1} - \mu_0(x_t,c,t)\|_2^2 \right]$$
$$= \mathbb{E}_{x_T \sim q_{0,t}(\cdot|x_t,c)} \left[ \mathbb{E}_{x_{t+1} \sim q_{0,t}(\cdot|x_t,x_T,c)} \left[ \|x_{t+1} - \mu_0(x_t,c,t)\|_2^2 \right] \right]. \tag{103}$$

By (112) we can compute the inner expectation as

$$\mathbb{E}_{x_{t+1} \sim q_{0,t}(\cdot|x_t,x_T,c)} \left[ \|x_{t+1} - \mu_0(x_t,c,t)\|_2^2 \right] = \mathbb{E}_{\xi \sim \mathcal{N}(0,\mathbf{I}_d)} \left[ \left\| \widetilde{\mu}_t(x_t,x_T,t) - \mu_0(x_t,c,t) + \sqrt{\widetilde{\beta}_t}\xi \right\|_2^2 \right]$$
$$= \|\widetilde{\mu}_t(x_t,x_T,t) - \mu_0(x_t,c,t)\|_2^2 + d\widetilde{\beta}_t, \tag{104}$$

where

$$\widetilde{\beta}_t = \frac{\beta_t(1 - \overline{\alpha}_{t+1})}{1 - \overline{\alpha}_t}, \quad \widetilde{\mu}_t(x_t,x_T,t) = \frac{\sqrt{\alpha_t}(1 - \overline{\alpha}_{t+1})}{1 - \overline{\alpha}_t}x_t + \frac{\beta_t\sqrt{\overline{\alpha}_{t+1}}}{1 - \overline{\alpha}_t}x_T. \tag{105}$$

Plugging (104) back into (103), we have

$$(a) = \mathbb{E}_{x_T \sim q_{0,t}(\cdot|x_t,c)} \left[ \|\widetilde{\mu}_t(x_t,x_T,t) - \mu_0(x_t,c,t)\|_2^2 \right] + d\widetilde{\beta}_t$$
$$= \mathbb{E}_{x_T \sim q_{0,t}(\cdot|x_t,c)} \left[ \left\| \widetilde{\mu}_t(x_t,x_T,t) - \mathbb{E}_{x_T \sim q_{0,t}(\cdot|x_t,c)} \left[ \widetilde{\mu}_t(x_t,x_T,t) \right] \right\|_2^2 \right] + d\widetilde{\beta}_t$$
$$\stackrel{(105)}{=} \left( \frac{\beta_t\sqrt{\overline{\alpha}_{t+1}}}{1 - \overline{\alpha}_t} \right)^2 \mathbb{E}_{x_T \sim q_t(\cdot|x_t,c)} \left[ \|x_T - \mathbb{E}\left[x_T|x_t,c\right]\|_2^2 \right] + d\widetilde{\beta}_t, \tag{106}$$

where the second equality follows from Lemma 2 and (115) in its proof. Plugging (106) and (102) back into (101), we have

$$\|\delta_0(x_t, c, t)\|_2 \leqslant \left( \frac{\psi_{\max}}{\psi_{\min}} - 1 \right) \left( 1 + \sqrt{2} \frac{\psi_{\max}}{\psi_{\min}} \right) \sqrt{2\mathrm{TV}\left(q_{0,t}, p_{0,t}\right)}$$
$$\cdot \sqrt{\left( \frac{\beta_t\sqrt{\overline{\alpha}_{t+1}}}{1 - \overline{\alpha}_t} \right)^2 \mathbb{E}_{x_T \sim q_{0,t}(\cdot|x_t,c)} \left[ \|x_T - \mathbb{E}\left[x_T|x_t,c\right]\|_2^2 \right] + d(\widetilde{\beta}_t + \beta_t)}$$
$$\leqslant \left( \frac{\psi_{\max}}{\psi_{\min}} - 1 \right) \left( 1 + \sqrt{2} \frac{\psi_{\max}}{\psi_{\min}} \right) \sqrt{2\mathrm{TV}\left(q_{0,t}, p_{0,t}\right)} \sqrt{\left( \frac{\beta_t\sqrt{\overline{\alpha}_{t+1}}}{1 - \overline{\alpha}_t} \right)^2 + d(\widetilde{\beta}_t + \beta_t)}, \tag{107}$$

where the last inequality follows from our assumption that $\|x_T\|_\infty \leqslant 1$.

## B.4. Proof of Lemma 1

Note that

$$\left\| \epsilon\left(x_t, c, t\right) + \sqrt{1-\overline{\alpha}_t}\nabla_{x_t} \log q_t^\star\left(x_t|c\right)\right\|_2^2$$

$$= \left\| \epsilon\left(x_t, c, t\right) + \frac{\sqrt{1-\overline{\alpha}_t}}{q_t^\star\left(x_t|c\right)}\nabla_{x_t} \int p^\star\left(x_T|c\right) q_t(x_t|x_T)dx_T\right\|_2^2$$

$$= \left\| \epsilon\left(x_t, c, t\right)\right\|_2^2 + \frac{2\sqrt{1-\overline{\alpha}_t}}{q_t^\star\left(x_t|c\right)}\left\langle \epsilon\left(x_t, c, t\right), \nabla_{x_t} \int p^\star\left(x_T|c\right) q_t(x_t|x_T)dx_T\right\rangle + \mathsf{const}$$

$$= \left\| \epsilon\left(x_t, c, t\right)\right\|_2^2 \underbrace{\frac{1}{q_t^\star\left(x_t|c\right)} \int p^\star\left(x_T|c\right) q_t(x_t|x_T)dx_T}_{=1}$$

$$+ \frac{2\sqrt{1-\overline{\alpha}_t}}{q_t^\star\left(x_t|c\right)} \int p^\star\left(x_T|c\right) q_t(x_t|x_T) \left\langle \epsilon\left(x_t, c, t\right), \nabla_{x_t} \log q_t(x_t|x_T)\right\rangle dx_T + \mathsf{const}$$

$$= \frac{1}{q_t^\star\left(x_t|c\right)} \int p^\star\left(x_T|c\right) q_t(x_t|x_T) \left(\left\| \epsilon\left(x_t, c, t\right)\right\|_2^2 + 2\sqrt{1-\overline{\alpha}_t}\left\langle \epsilon\left(x_t, c, t\right), \nabla_{x_t} \log q_t(x_t|x_T)\right\rangle\right) dx_T + \mathsf{const}$$

$$= \frac{1}{q_t^\star\left(x_t|c\right)} \int p^\star\left(x_T|c\right) q_t(x_t|x_T) \left(\left\| \epsilon\left(x_t, c, t\right) + \sqrt{1-\overline{\alpha}_t}\nabla_{x_t} \log q_t(x_t|x_T)\right\|_2^2\right) dx_T + \mathsf{const}, \tag{108}$$

where const denotes constants that are independent of $\epsilon$.

Integrate both sides of the above equation over $x_t \sim q_t^\star\left(\cdot|c\right)$, we have

$$\mathbb{E}_{x_t \sim q_t^\star(\cdot|c)} \left[\left\| \epsilon\left(x_t, c, t\right) + \sqrt{1-\overline{\alpha}_t}\nabla_{x_t} \log q_t^\star\left(x_t|c\right)\right\|_2^2\right]$$

$$= \mathbb{E}_{\substack{x_T \sim p^\star(\cdot|c), \\ x_t \sim q_t(\cdot|x_T)}} \left[\left\| \epsilon\left(x_t, c, t\right) + \sqrt{1-\overline{\alpha}_t}\nabla_{x_t} \log q_t(x_t|x_T)\right\|_2^2\right] + \mathsf{const}$$

$$= \mathbb{E}_{\substack{x_T \sim p^\star(\cdot|c), \\ \xi \sim \mathcal{N}(0, \mathbf{I}_d)}} \left[\left\| \xi - \epsilon\left(\sqrt{\overline{\alpha}_t}x_T + \sqrt{1-\overline{\alpha}_t}\xi, c, t\right)\right\|_2^2\right] + \mathsf{const}. \tag{109}$$

By comparing (109) with $\mathcal{L}_{\mathrm{org}}\left(\cdot; p^\star\right)$, we obtain

$$\mathcal{L}_{\mathrm{org}}\left(\epsilon; p^\star\right) = \frac{1}{2}\mathbb{E}_{\substack{c \sim p_c, t \sim \mathcal{U}([T-1]), \\ x_t \sim q_t^\star(\cdot|c)}} \left[\left\| \epsilon\left(x_t, c, t\right) + \sqrt{1-\overline{\alpha}_t}\nabla_{x_t} \log q_t^\star\left(x_t|c\right)\right\|_2^2\right] + \mathsf{const}. \tag{110}$$

Recall we assume $\mathrm{supp}\left(p_c\right) = \mathcal{C}$. Therefore, (110) indicates the minimum of $\mathcal{L}_{\mathrm{org}}\left(\cdot; p^\star\right)$ is $\epsilon_{p^\star}$ that satisfies (40). On the other hand, it's obvious that for any $g : \mathbb{R}^d \times \mathcal{C} \times [T] \to (0, +\infty)$, the minimum of $\mathcal{L}$ is also $\epsilon_{p^\star}$, as long as (39) is well-defined.

## B.5. Proof of Lemma 2

We let

$$\mu_q(x_t, c, t) := \mathbb{E}_{x_{t+1} \sim q_t(\cdot|x_t, c)}\left[x_{t+1}\right]. \tag{111}$$

First note that by the forward process rule (4) and Bayes' rule, we have

$$q_t(x_{t+1}|x_t, x_T) = \frac{q_{t+1}(x_{t+1}|x_T)p_t(x_t|x_{t+1})}{q_t(x_t|x_T)} = \mathcal{N}\left(x_{t+1}|\widetilde{\mu}_t(x_t, x_T, t), \widetilde{\beta}_t\mathbf{I}_d\right) \tag{112}$$

with

$$\widetilde{\mu}_t(x_t, x_T, t) = \frac{\sqrt{\alpha_t}(1-\overline{\alpha}_{t+1})}{1-\overline{\alpha}_t}x_t + \frac{\beta_t\sqrt{\overline{\alpha}_{t+1}}}{1-\overline{\alpha}_t}x_T \tag{113}$$

and

$$\widetilde{\beta}_t = \frac{\beta_t(1 - \overline{\alpha}_{t+1})}{1 - \overline{\alpha}_t}. \tag{114}$$

Thus we have

$$\mu_q(x_t, c, t) = \mathbb{E}_{x_T \sim q_t(\cdot | x_t, c)} \left[ \widetilde{\mu}_t(x_t, x_T, t) \right] = \frac{\sqrt{\alpha_t}(1 - \overline{\alpha}_{t+1})}{1 - \overline{\alpha}_t} x_t + \frac{\beta_t \sqrt{\overline{\alpha}_{t+1}}}{1 - \overline{\alpha}_t} \mathbb{E}_{x_T \sim q_t(\cdot | x_t, c)} \left[ x_T \right]. \tag{115}$$

On the other hand, since $\epsilon_p^\star$ minimizes the noise-prediction regression objective $\mathcal{L}_{\mathrm{org}}(\cdot; p)$ (c.f. (10)), we have

$$\epsilon_p^\star(x_t, c, t) = \mathbb{E}[\xi | x_t, c, t] \overset{(4)}{=} \mathbb{E} \left[ \frac{x_t - \sqrt{\overline{\alpha}_t} x_T}{\sqrt{1 - \overline{\alpha}_t}} \,\middle|\, x_t, c, t \right] = \frac{x_t}{\sqrt{1 - \overline{\alpha}_t}} - \frac{\sqrt{\overline{\alpha}_t}}{\sqrt{1 - \overline{\alpha}_t}} \mathbb{E}_{x_T \sim q_t(\cdot | x_t, c)} \left[ x_T \right], \tag{116}$$

from which we deduce

$$
\begin{aligned}
\mu_p^\star(x_t, c, t) &= \frac{1}{\sqrt{\alpha_t}} x_t - \frac{\beta_t}{\sqrt{\alpha_t (1 - \overline{\alpha}_t)}} \epsilon_p^\star(x_t, c, t) \\
&= \frac{1}{\sqrt{\alpha_t}} x_t - \frac{\beta_t}{\sqrt{\alpha_t (1 - \overline{\alpha}_t)}} \left( \frac{x_t}{\sqrt{1 - \overline{\alpha}_t}} - \frac{\sqrt{\overline{\alpha}_t}}{\sqrt{1 - \overline{\alpha}_t}} \mathbb{E}_{x_T \sim q_t(\cdot | x_t, c)} \left[ x_T \right] \right) \\
&= \frac{\sqrt{\alpha_t}(1 - \overline{\alpha}_{t+1})}{1 - \overline{\alpha}_t} x_t + \frac{\beta_t \sqrt{\overline{\alpha}_{t+1}}}{1 - \overline{\alpha}_t} \mathbb{E}_{x_T \sim q_t(\cdot | x_t, c)} \left[ x_T \right] \overset{(115)}{=} \mu_q(x_t, c, t).
\end{aligned}
\tag{117}
$$

# C. Additional Experiment Details

**Details on evaluation.** For automatic evaluation, we found at test time, enlarging the side network guidance strength $\lambda_{\text{model}}$ can help the finetuned models to achieve better performance, especially for SFT and RWL. Thus for SFT and RWL, we evaluate DiffCon, DiffCon-J, DiffCon-S with $1/\lambda_{\text{model}} \in \{0.1, 0.2, \cdots, 1.0\}$ and DiffCon-Naive with $\lambda_{\text{model}} \in \{1.0, 2.0, \cdots, 10.0\}$; for PPO, we evaluate DiffCon, DiffCon-J, DiffCon-S with $1/\lambda_{\text{model}} \in \{0.7, 0.8, 0.9, 1.0\}$ and DiffCon-Naive with $\lambda_{\text{model}} \in \{1.0, 2.0, 3.0, 4.0\}$, and we report the best HPS-v2 win rate among them for each checkpoint.

For human evaluation, we recruited more than 50 human raters [3] to rate the generated images of three pairs of network structures trained with PPO: (i) (gray-box) DiffCon vs. DiffCon-Naive, (ii) (white-box) DiffCon-J vs. LoRA, (iii) (white-box) DiffCon-S vs. LoRA. We randomly generate 50 prompts from the HPS-v2 prompt set. For each network structure we generate four images using different seeds for each prompt. The raters are tasked to evaluate a uniformly sampled pair of network structures out of the three, and are instructed to pick their favorite set based on the best-of-four to do evaluation.

**DiffCon parameterization.** We instantiate the side network $s_\theta = (z_{\theta_1}, h_{\theta_2})$ for DiffCon as a lightweight UNet operating on the latent space (4+1 channels at $64 \times 64$ for SD v1.4). It consists of standard UNet blocks (`DownBlock2D`, `CrossAttnDownBlock2D` for the encoder, `UpBlock2D`, `CrossAttnUpBlock2D` for the decode, and `CrossAttnMidBlock2D` in the middle) similar as the ones used in `diffusers` (von Platen et al., 2022). See Table 2 for the details of the blocks.

We found that using two `DownBlock2D` for the encoder and two `UpBlock2D` for the decoder works best, and the results presented in the main paper are under this structure. See Table 3 for the details of our side network structure under this choice. In our ablation study, we also tried other structures, see Appendix D for more details.

*Table 2.* Block definitions used in $s_\theta$. **Common components:** `ResnetBlock2D`: Standard ResNet block with GroupNorm, SiLU, and two Convolutional layers with kernel size $3 \times 3$ and stride 1. `Transformer2DModel`: spatial transformer with cross-attention. `Downsample2D`: a convolutional layer with kernel size $3 \times 3$ and stride 2. `Upsample2D`: $2\times$ Nearest Neighbor Resize + $3 \times 3$ conv, stride 1.

| Block Name | Structure & Sequence |
|:---:|:---:|
| `CrossAttnDownBlock2D` | `ResnetBlock2D`
`Transformer2DModel`
`Downsample2D` (optional) |
| `DownBlock2D` | `ResnetBlock2D`
`Downsample2D` (optional) |
| `CrossAttnMidBlock2D` | `ResnetBlock2D`
`Transformer2DModel`
`ResnetBlock2D` |
| `CrossAttnUpBlock2D` | `ResnetBlock2D`
`Transformer2DModel`
`Upsample2D` (optional) |
| `UpBlock2D` | `ResnetBlock2D`
`Upsample2D` (optional) |

$s_\theta$ **(DiffCon) vs.** $\bar{s}_\theta$ **(DiffCon-Naive).** Both DiffCon and DiffCon-Naive use the same lightweight UNet-style side network backbone, but they differ in (i) what is fed into the side network and (ii) how its output is used. In DiffCon, the side network is *structured* as $s_\theta = (z_{\theta_1}, h_{\theta_2})$ and is evaluated on the pretrained reverse mean $\mu_0(x_t, c, t)$ (c.f. (27), (28)); one output head produces a scalar gate $z_{\theta_1}$ (implemented as an extra output channel followed by a linear head), while the remaining channels form $h_{\theta_2}$. In contrast, DiffCon-Naive uses an almost identical backbone but removes this gating head (one fewer output channel) and does not split outputs; it directly predicts an additive residual that is added to $\epsilon_0$. We provide a detailed comparison in Table 4.

---

[3]The raters were paid contractors. They received their standard contracted wage, which is above the living wage in their country of employment.

*Table 3.* Side network structure used in the main paper.

**Configuration:** `layers_per_block=1`, `block_out_channels=(128, 256)`, `sample_size=64 × 64`, `down_block_types=(DownBlock2D, DownBlock2D)`, `up_block_types=(UpBlock2D, UpBlock2D)`.

| Stage | Block Sequence | Input Channels | Output Channels | Resolution | Details |
|---|---|---|---|---|---|
| Input conv | Conv2D | 4 | 128 | $64 \times 64 \to 64 \times 64$ | ker $3 \times 3$, stride 1. |
| Down block 0 | DownBlock2D | 128 | 128 | $64 \times 64 \to 32 \times 32$ | Downsample: True. |
| Down block 1 | DownBlock2D | 128 | 256 | $32 \times 32 \to 32 \times 32$ | Downsample: False. |
| Mid block | CrossAttnMidBlock2D256 | 256 | 256 | $32 \times 32 \to 32 \times 32$ | - |
| Up block 0 | UpBlock2D | 512 (256+256) | 256 | $32 \times 32 \to 64 \times 64$ | Skip from Down block 1. Upsample: True. |
| Up block 1 | UpBlock2D | 384 (256+128) | 128 | $64 \times 64 \to 64 \times 64$ | Skip from Down block 0. Upsample: False. |
| Output conv | Conv2D | 128 | 4 | $64 \times 64 \to 64 \times 64$ | ker $3 \times 3$, stride 1. |

| | **DiffCon:** $s_\theta$ | **DiffCon-Naive:** $\overline{s}_\theta$ |
|---|---|---|
| Backbone | lightweight UNet | same backbone (by design) |
| Side-net input | $(\mu_0(x_t, c, t), c, t)$ | $(x_t, c, t)$ |
| Output parameterization | $(z_{\theta_1}, h_{\theta_2})$ | single tensor (no split) |
| Output channels (last conv) | 5 (extra channel for $z_{\theta_1}$) | 4 (one fewer channel) |
| How output is used | controller-form correction (27), (28) | direct residual: $\epsilon_\theta = \epsilon_0 + \lambda_{\text{model}}\overline{s}_\theta$ |
| Intuition | gated, structured control term | ungated additive correction |

*Table 4.* Side network variants in gray-box finetuning. Both methods share the same side-net backbone, but DiffCon uses a structured output $(z, h)$ applied in the controller-form correction, while DiffCon-Naive directly adds an unsplit residual.

## D. Additional Experiment Results

To meet the ICML paper-size requirement, images in this section are compressed after compilation.

**Example generations.** Table 5, 6, 7 shows some generation examples for each method under the SFT, RWL, and PPO settings. In addition, Figure 4, 5, 6, 7 show some generated images with varying guidance strength $\lambda_{\text{model}}$ for both DiffCon and DiffCon-J methods under the SFT and RWL settings, demonstrating the effectiveness of guidance at inference time.

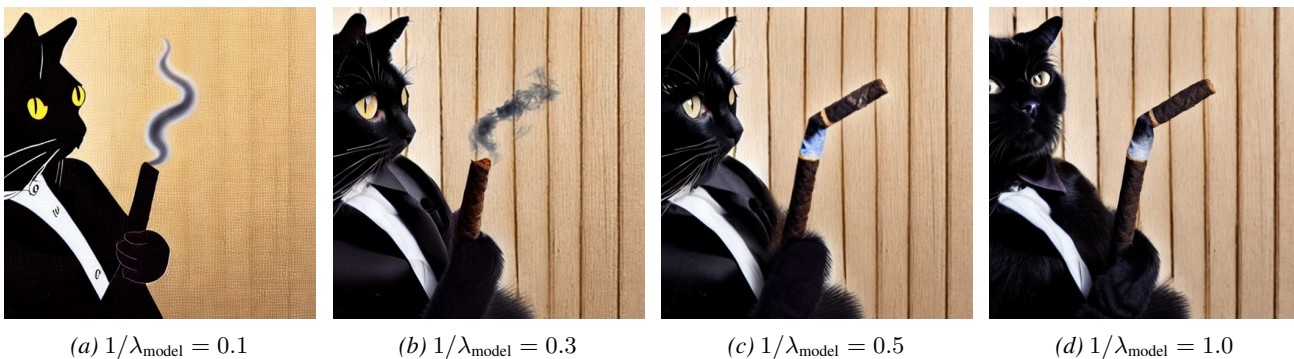

*(a)* $1/\lambda_{\text{model}} = 0.1$     *(b)* $1/\lambda_{\text{model}} = 0.3$     *(c)* $1/\lambda_{\text{model}} = 0.5$     *(d)* $1/\lambda_{\text{model}} = 1.0$

*Figure 4.* Generations with different guidance strengths $\lambda_{\text{model}}$ using SFT with DiffCon. Prompt: "A black cat wearing a suit and smoking a cigar".

**Full automatic evaluation metrics.** We report the HPS-v2 win rate vs. the pretrained model together with CLIP, CLIP-Aesthetics, and PickScore on the HPS-v2 test prompt set, for each algorithm at its reported checkpoint. We can see that HPS-v2 is improved by our methods, and the metrics are not compromised during our finetuning.

**Our RWL v.s. DPOK (Fan et al., 2023, Algorithm 2).** We compare the heuristic reward weight loss in DPOK (Fan et al., 2023, Algorithm 2) and our proposed polynomial reward weighting (21) using LoRA in Figure 8. For DPOK, we set

*Table 5.* Qualitative comparison of different methods using SFT training. We compare Pretrained sd v1.4, DiffCon, DiffCon-Naive, LoRA, DiffCon-J, and DiffCon-S.

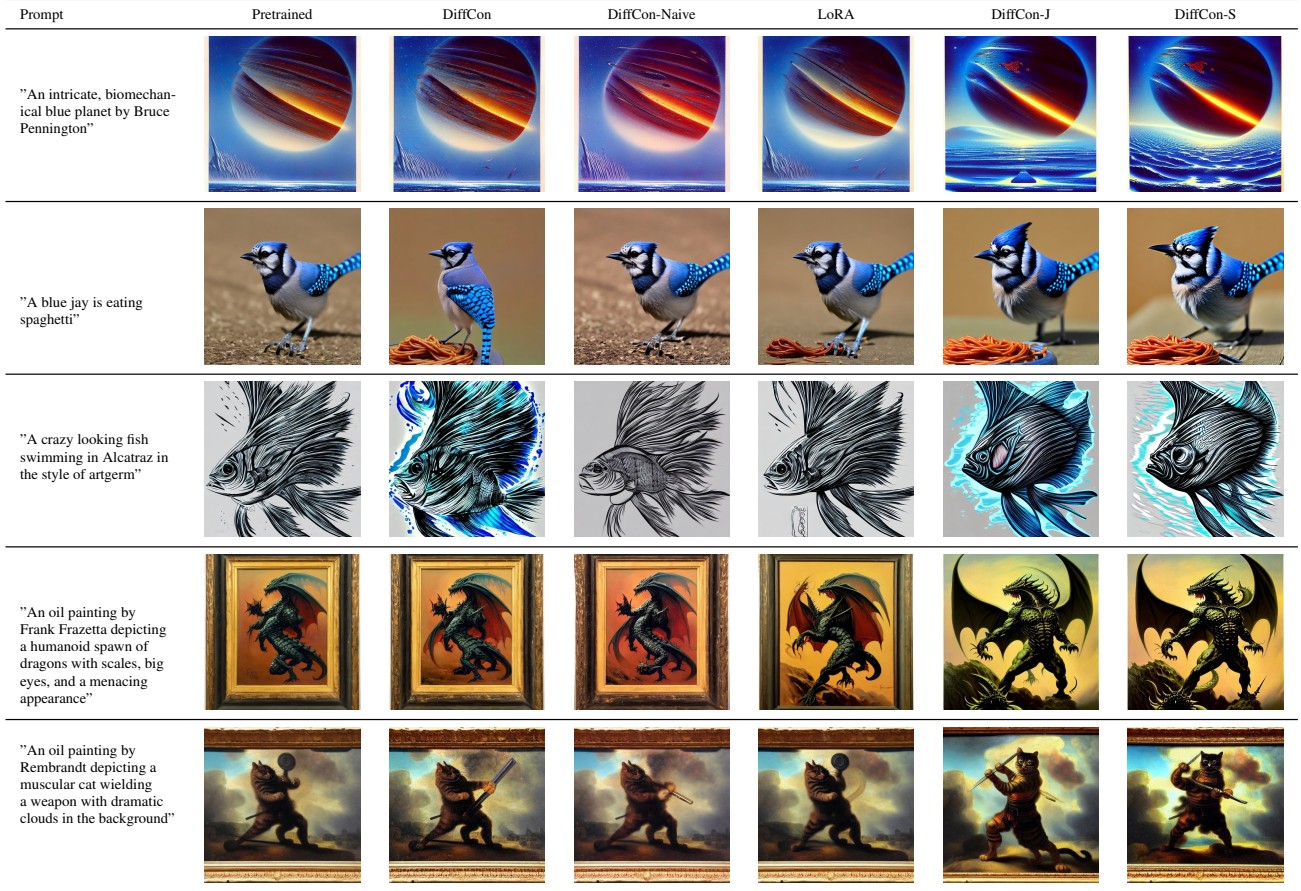

| Prompt | Pretrained | DiffCon | DiffCon-Naive | LoRA | DiffCon-J | DiffCon-S |
|---|---|---|---|---|---|---|
| "An intricate, biomechanical blue planet by Bruce Pennington" | | | | | | |
| "A blue jay is eating spaghetti" | | | | | | |
| "A crazy looking fish swimming in Alcatraz in the style of artgerm" | | | | | | |
| "An oil painting by Frank Frazetta depicting a humanoid spawn of dragons with scales, big eyes, and a menacing appearance" | | | | | | |
| "An oil painting by Rembrandt depicting a muscular cat wielding a weapon with dramatic clouds in the background" | | | | | | |

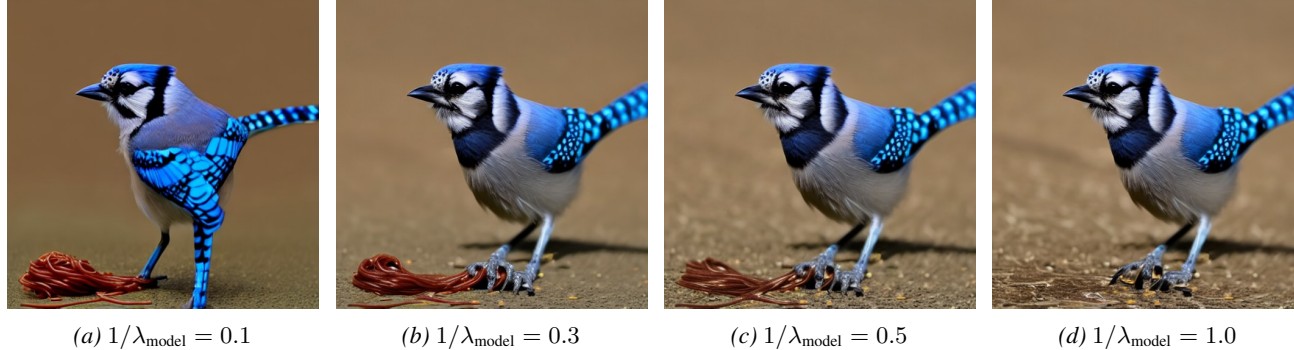

(a) $1/\lambda_{\mathrm{model}} = 0.1$    (b) $1/\lambda_{\mathrm{model}} = 0.3$    (c) $1/\lambda_{\mathrm{model}} = 0.5$    (d) $1/\lambda_{\mathrm{model}} = 1.0$

*Figure 5.* Generations with different guidance strengths $\lambda_{\mathrm{model}}$ using SFT with DiffCon-J. Prompt: "A blue jay is eating spaghetti".

*Table 6.* Qualitative comparison of different methods using RWL training. We compare Pretrained sd v1.4, DiffCon, DiffCon-Naive, LoRA, DiffCon-J, and DiffCon-S.

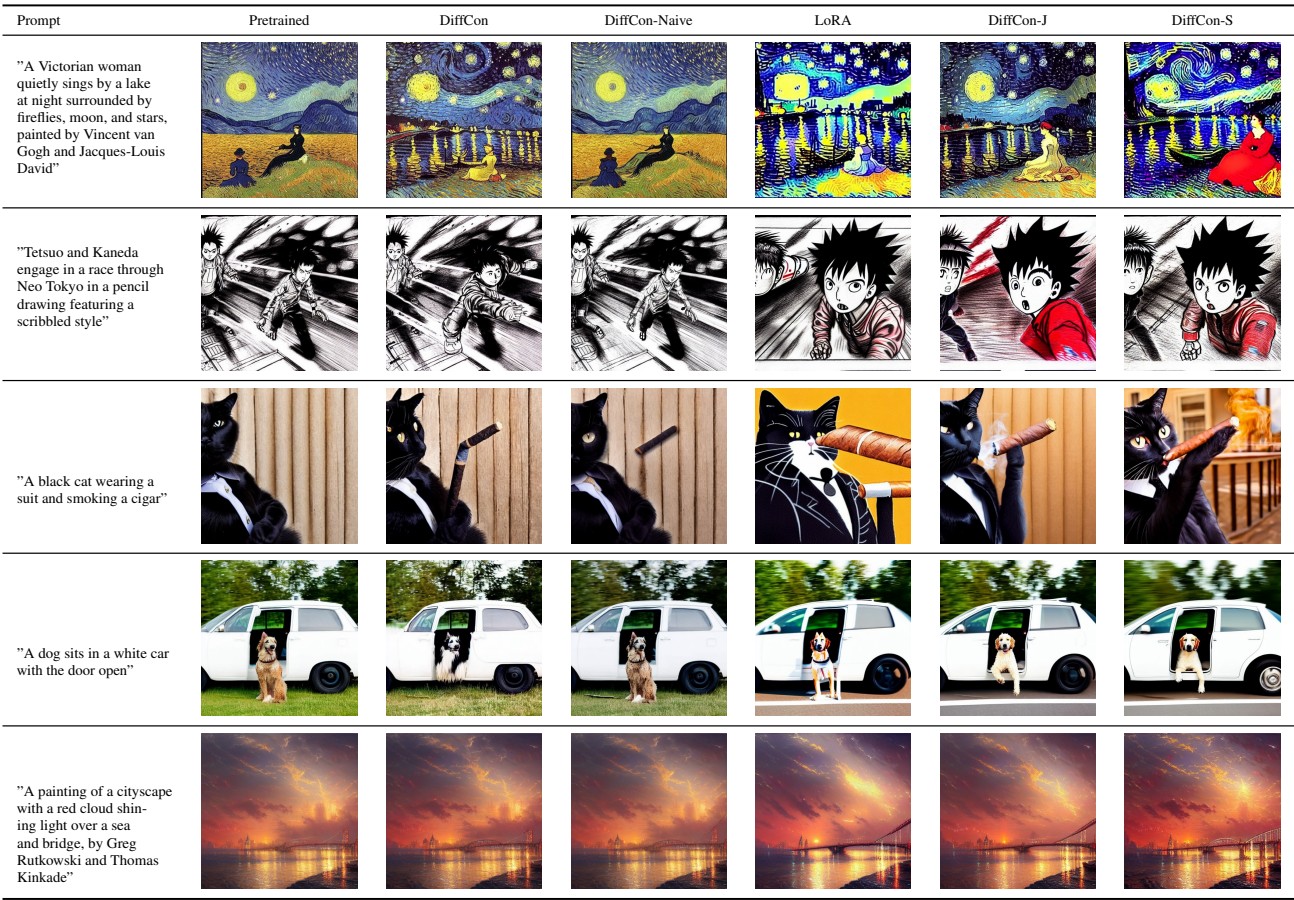

| Prompt | Pretrained | DiffCon | DiffCon-Naive | LoRA | DiffCon-J | DiffCon-S |
|---|---|---|---|---|---|---|

"A Victorian woman quietly sings by a lake at night surrounded by fireflies, moon, and stars, painted by Vincent van Gogh and Jacques-Louis David"

"Tetsuo and Kaneda engage in a race through Neo Tokyo in a pencil drawing featuring a scribbled style"

"A black cat wearing a suit and smoking a cigar"

"A dog sits in a white car with the door open"

"A painting of a cityscape with a red cloud shining light over a sea and bridge, by Greg Rutkowski and Thomas Kinkade"

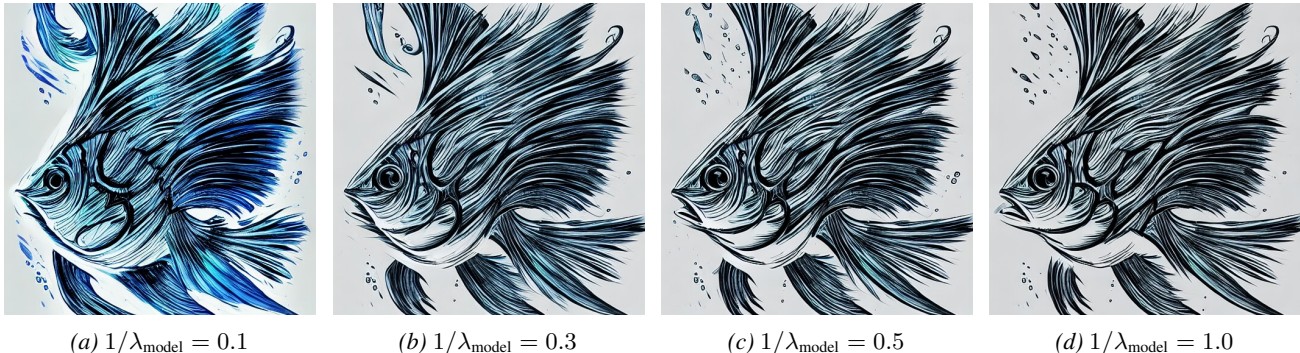

*(a)* $1/\lambda_{\mathrm{model}} = 0.1$      *(b)* $1/\lambda_{\mathrm{model}} = 0.3$      *(c)* $1/\lambda_{\mathrm{model}} = 0.5$      *(d)* $1/\lambda_{\mathrm{model}} = 1.0$

*Figure 6.* Generations with different guidance strengths $\lambda_{\mathrm{model}}$ using RWL with DiffCon. Prompt: "A crazy looking fish swimming in Alcatraz in the style of artgerm".

*Table 7.* Qualitative comparison of different methods using PPO training. We compare Pretrained sd v1.4, DiffCon, DiffCon-Naive, LoRA, DiffCon-J, and DiffCon-S.

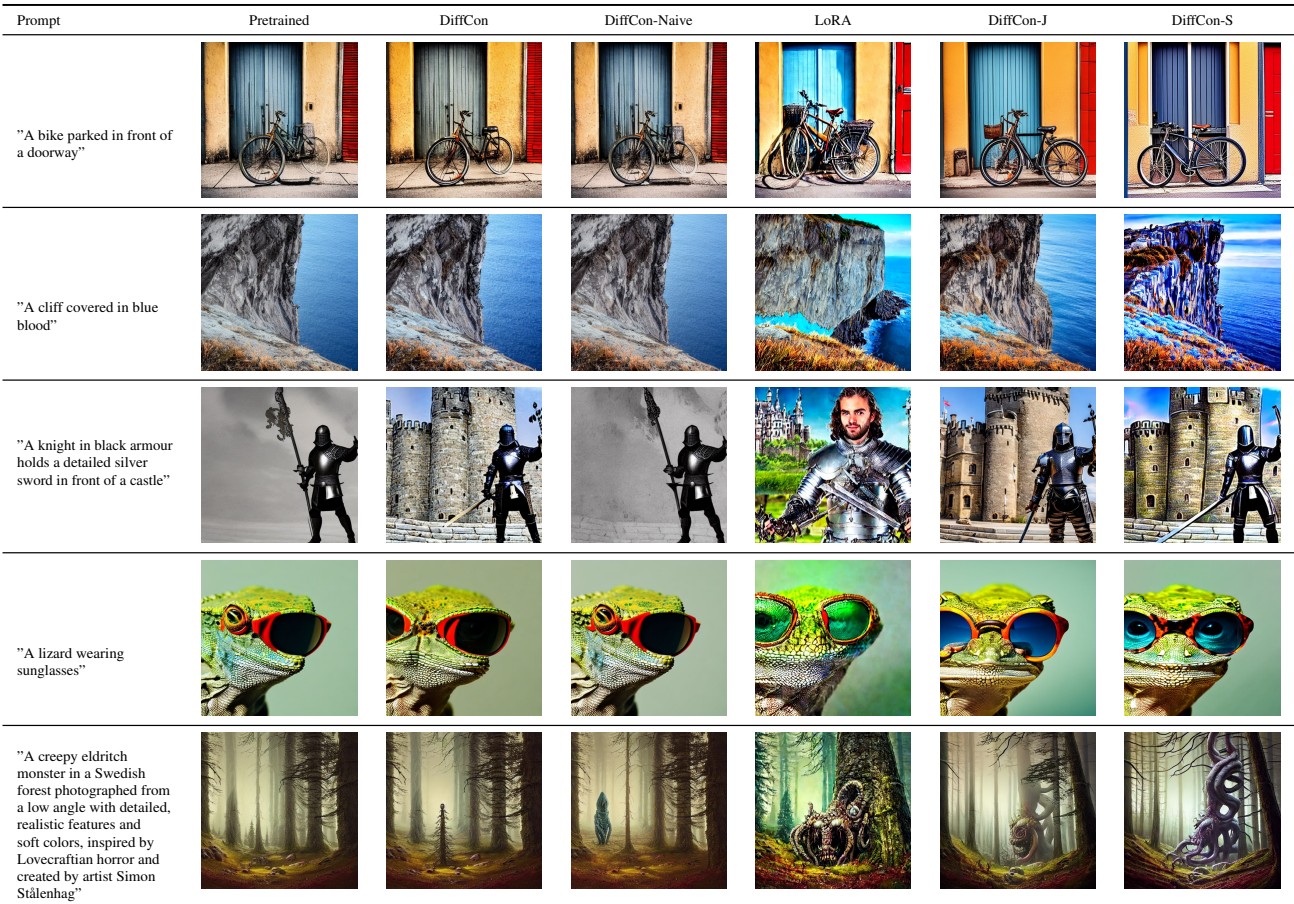

| Prompt | Pretrained | DiffCon | DiffCon-Naive | LoRA | DiffCon-J | DiffCon-S |
|---|---|---|---|---|---|---|
| "A bike parked in front of a doorway" | | | | | | |
| "A cliff covered in blue blood" | | | | | | |
| "A knight in black armour holds a detailed silver sword in front of a castle" | | | | | | |
| "A lizard wearing sunglasses" | | | | | | |
| "A creepy eldritch monster in a Swedish forest photographed from a low angle with detailed, realistic features and soft colors, inspired by Lovecraftian horror and created by artist Simon Stålenhag" | | | | | | |

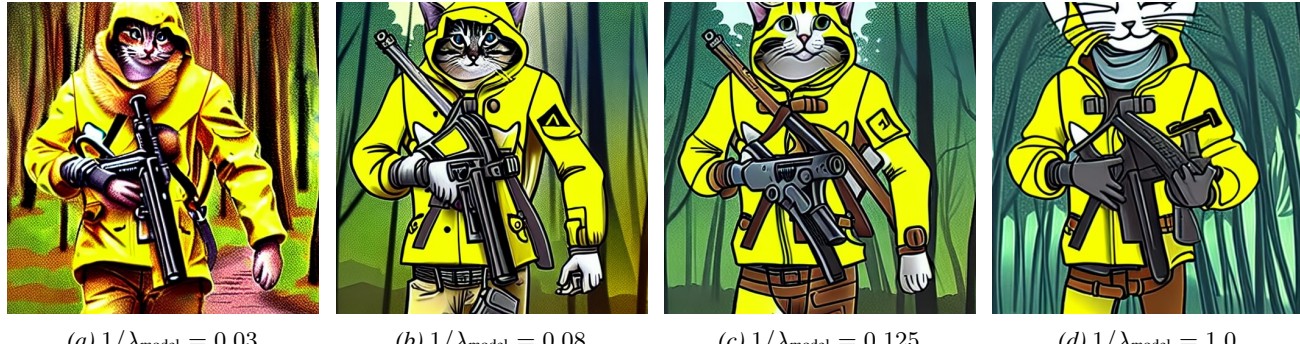

(a) $1/\lambda_{\text{model}} = 0.03$     (b) $1/\lambda_{\text{model}} = 0.08$     (c) $1/\lambda_{\text{model}} = 0.125$     (d) $1/\lambda_{\text{model}} = 1.0$

*Figure 7.* Generations with different guidance strengths $\lambda_{\text{model}}$ using RWL with DiffCon-J. Prompt: "A cute humanoid cat soldier wears a yellow raincoat, carries a rifle, and ventures through a dense forest, looking back over their shoulder".

| Method | # Params ↓ | Gray-box | HPS-v2 win rate vs. pretrained model ↑ | CLIP ↑ | CLIP-Aesthetics ↑ | PickScore ↑ |
|---|---|---|---|---|---|---|
| Pretrained | - | - | 0.500 | 0.2750 | 5.5163 | 0.2103 |
| **DiffCon (ours)** | $1.2 \times 10^7$ | ✓ | **0.6667 ± 0.0028** | 0.2744 ± 0.0004 | 5.4754 ± 0.0325 | 0.2105 ± 0.0001 |
| DiffCon-Naive | $1.2 \times 10^7$ | ✓ | 0.5655 ± 0.0165 | 0.2740 ± 0.0002 | 5.5153 ± 0.0048 | 0.2105 ± 0.0001 |
| LoRA | $1.7 \times 10^7$ | ✗ | 0.5766 ± 0.0137 | 0.2760 ± 0.0002 | 5.5144 ± 0.0014 | 0.2107 ± 0.0000 |
| **DiffCon-J (ours)** | $1.6 \times 10^7$ | ✗ | **0.6964 ± 0.0293** | 0.2747 ± 0.0002 | 5.4531 ± 0.0013 | 0.2103 ± 0.0001 |
| **DiffCon-S (ours)** | $1.6 \times 10^7$ | ✗ | **0.6964 ± 0.0018** | 0.2747 ± 0.0010 | 5.4678 ± 0.0245 | 0.2104 ± 0.0003 |

*Table 8.* SFT evaluation results on HPS-v2 test prompt set at step 1000.

| Method | # Params ↓ | Gray-box | HPS-v2 win rate vs. pretrained model ↑ | CLIP ↑ | CLIP-Aesthetics ↑ | PickScore ↑ |
|---|---|---|---|---|---|---|
| Pretrained | - | - | 0.500 | 0.2750 | 5.5163 | 0.2103 |
| **DiffCon (ours)** | $1.2 \times 10^7$ | ✓ | **0.6815 ± 0.0111** | 0.2732 ± 0.0009 | 5.4448 ± 0.0265 | 0.2105 ± 0.0001 |
| DiffCon-Naive | $1.2 \times 10^7$ | ✓ | 0.5060 ± 0.0190 | 0.2736 ± 0.0008 | 5.4977 ± 0.0118 | 0.2101 ± 0.0000 |
| LoRA | $1.7 \times 10^7$ | ✗ | 0.6109 ± 0.0059 | 0.2696 ± 0.0007 | 5.4950 ± 0.0069 | 0.2096 ± 0.0003 |
| **DiffCon-J (ours)** | $1.6 \times 10^7$ | ✗ | **0.6555 ± 0.0210** | 0.2712 ± 0.0007 | 5.5037 ± 0.0294 | 0.2100 ± 0.0004 |
| **DiffCon-S (ours)** | $1.6 \times 10^7$ | ✗ | **0.7091 ± 0.0155** | 0.2710 ± 0.0010 | 5.4714 ± 0.0311 | 0.2102 ± 0.0004 |

*Table 9.* RWL evaluation results on HPS-v2 test prompt set at step 2000.

| Method | # Params ↓ | Gray-box | HPS-v2 win rate vs. pretrained model ↑ | CLIP ↑ | CLIP-Aesthetics ↑ | PickScore ↑ |
|---|---|---|---|---|---|---|
| Pretrained | - | - | 0.500 | 0.2750 | 5.5163 | 0.2103 |
| **DiffCon (ours)** | $1.2 \times 10^7$ | ✓ | **0.6957 ± 0.0152** | 0.2711 ± 0.0004 | 5.3370 ± 0.0219 | 0.2094 ± 0.0001 |
| DiffCon-Naive | $1.2 \times 10^7$ | ✓ | 0.5201 ± 0.0203 | 0.2748 ± 0.0006 | 5.5186 ± 0.0079 | 0.2106 ± 0.0000 |
| LoRA | $1.7 \times 10^7$ | ✗ | 0.9048 ± 0.0084 | 0.2642 ± 0.0026 | 5.5631 ± 0.0642 | 0.2133 ± 0.0002 |
| **DiffCon-J (ours)** | $1.6 \times 10^7$ | ✗ | **0.9353 ± 0.0079** | 0.2660 ± 0.0004 | 5.6292 ± 0.0657 | 0.2142 ± 0.0007 |
| **DiffCon-S (ours)** | $1.6 \times 10^7$ | ✗ | **0.9315 ± 0.0164** | 0.2635 ± 0.0028 | 5.4989 ± 0.0846 | 0.2125 ± 0.0012 |

*Table 10.* PPO evaluation results on HPS-v2 test prompt set at step 2400.

the learning rate to be 3e-6, the batch size to be 128 (twice of ours) and sample images (from the pretrained model for the teacher sampling setting, and from the current diffusion policy for the online sampling setting) every iteration (the sample frequency is also twice of ours). The other configurations for DPOK are the same as our RWL for LoRA. We can see that our proposed RWL achieves better HPS-v2 win rate than DPOK with 1/4 sample complexity.

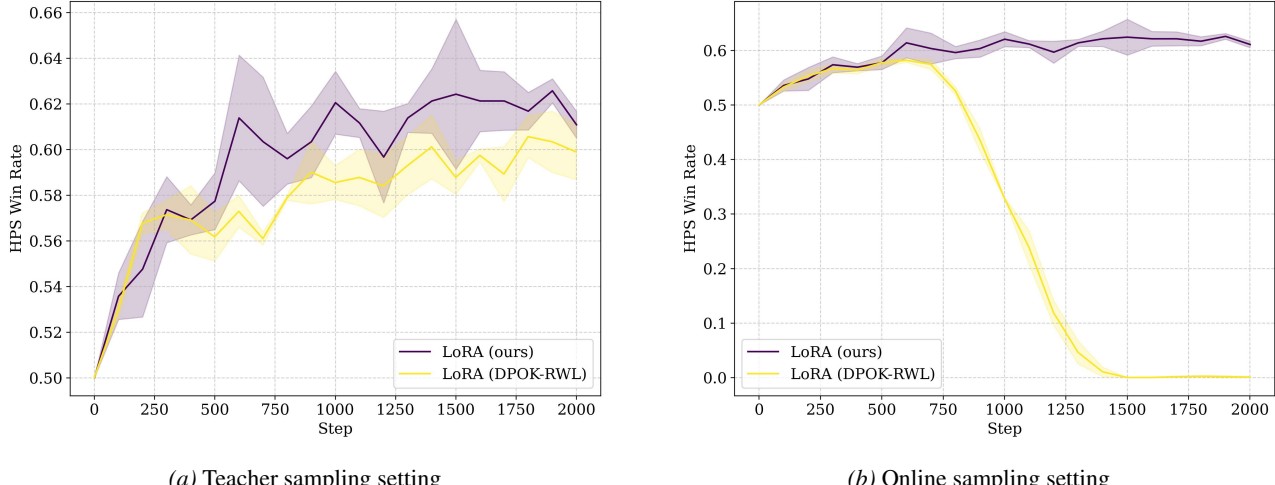

*(a)* Teacher sampling setting        *(b)* Online sampling setting

*Figure 8.* HPS-v2 win rate comparison between our polynomial reward weighting (LoRA, ours) and DPOK-RWL (LoRA) during RWL finetuning. (a) uses teacher sampling where the images for training at each iteration are generated by the pretrained model, and (b) uses online sampling where the images for training are generated by the current diffusion policy.

**Ablation studies on learning rates for RL algorithms.** We found LoRA performs better with higher learning rate (1e-4) for RWL and PPO, while DiffCon performs better with a lower learning rate (1e-5), as measured by HPS win rate (Figure 9).

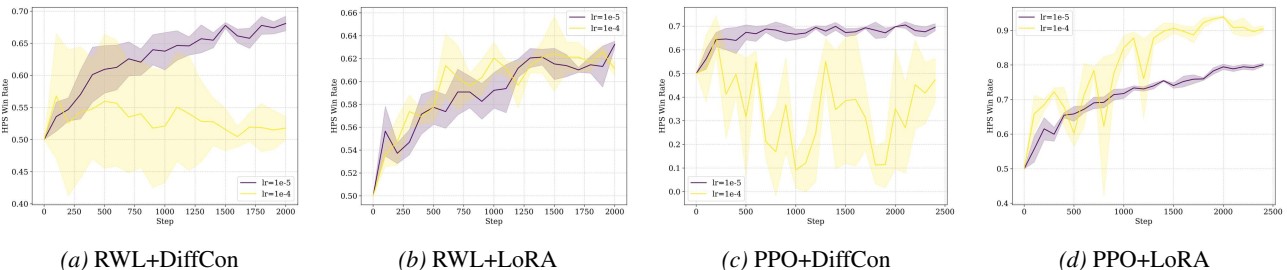

*(a)* RWL+DiffCon      *(b)* RWL+LoRA      *(c)* PPO+DiffCon      *(d)* PPO+LoRA

*Figure 9.* Learning rate ablation for RWL and PPO on HPS-v2 win rate (comparing 1e-4 vs. 1e-5).

**Ablation studies on $\tau_{\mathrm{KL}}$ for PPO.** We provide the effectiveness of different $\tau_{\mathrm{KL}}$ for PPO in Figure 10 to justify our choice of $\tau_{\mathrm{KL}}$ =1e-4 in the main paper, which leads to the best performance. Note that when the KL regularization is too small (or is 0), the reward won't go up.

**Ablation studies on $\tau_{\mathrm{RWL}}$ and weighting functions for RWL.** In the main paper, we report RWL results using the polynomial reward weighting function $w_\alpha$ (Eq. (21)). We first ablate $\tau_{\mathrm{RWL}}$ under this polynomial weighting: smaller $\tau_{\mathrm{RWL}}$ generally leads to better performance (Figure 11), but too small $\tau_{\mathrm{RWL}}$ (smaller than 0.0005) will cause instability in training (NaN in reward weight computation). Thus we choose $\tau_{\mathrm{RWL}} = 0.0005$ in the main paper.

We additionally report RWL results under the exponential reward weighting function $w_{\mathrm{KL}}$ (Eq. (20)) with $\tau_{\mathrm{RWL}} = 0.0015$ in Figure 12. Consistent with our main results, our methods achieve higher HPS win rate than the baselines, and DiffCon (with fewer trainable parameters) outperforms the strong white-box LoRA baseline.

**Ablation studies on side network structures.** As mentioned in Appendix C, we found that using two `DownBlock2D` for the encoder and two `UpBlock2D` for the decoder (i.e., down_block_types=(DownBlock2D, DownBlock2D),

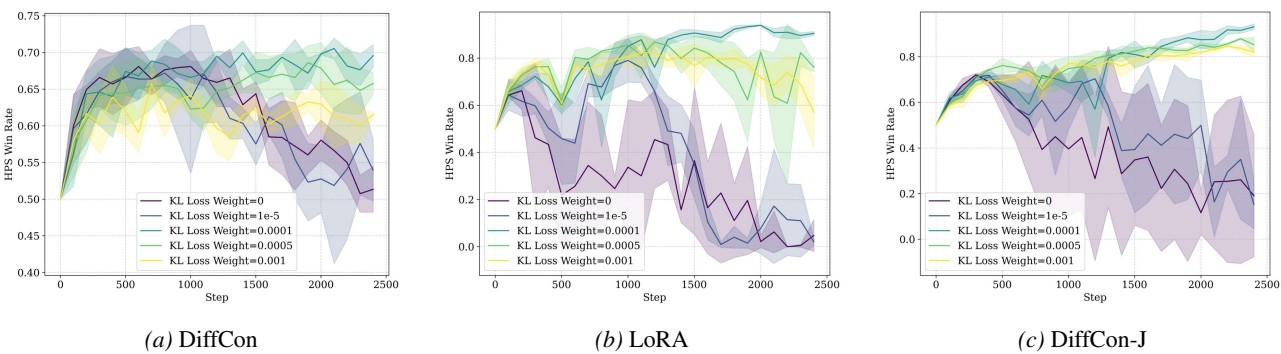

*(a)* DiffCon     *(b)* LoRA     *(c)* DiffCon-J

*Figure 10.* Ablation studies on $\tau_{KL}$ for PPO on HPS-v2 win rate (five learning curves correspond to $\tau_{KL} \in \{0, 1e\text{-}5, 1e\text{-}4, 5e\text{-}4, 1e\text{-}3\}$). $\tau_{KL} =$1e-4 leads to the best performance.

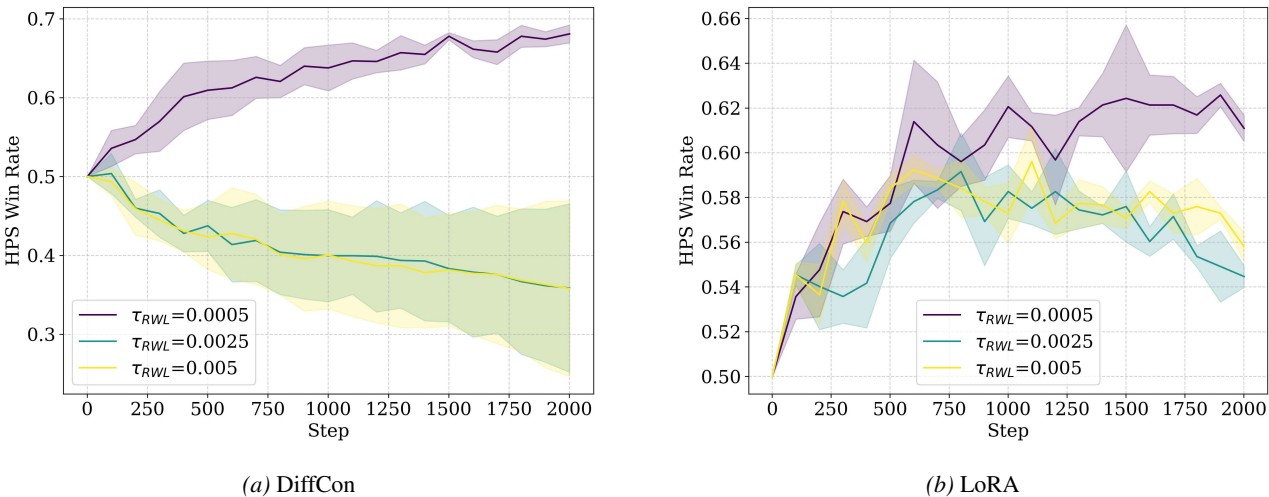

*(a)* DiffCon         *(b)* LoRA

*Figure 11.* Ablation studies on $\tau_{RWL}$ for RWL (three HPS-v2 win rate curves in each subfigure correspond to $\tau_{RWL} \in \{0.0005, 0.0025, 0.005\}$). $\tau_{RWL} = 0.0005$ leads to the best performance.

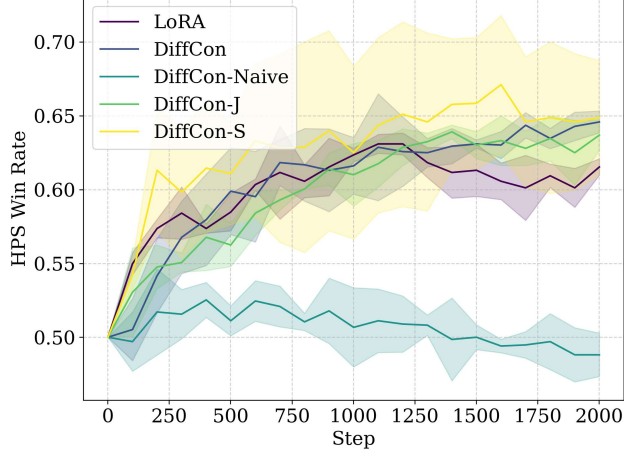

*Figure 12.* RWL under the exponential reward weighting function $w_{KL}$ (Eq. (20)) with $\tau_{RWL} = 0.0015$ on HPS-v2 win rate.

up_block_types=(UpBlock2D, UpBlock2D)) works best. We also tried different side network block combinations; see Table 11 and Figure 13 for more details.

*Table 11.* Side network structure variants used in the ablation study.

| Label | # trainable params | down_block_types | up_block_types | block_out_channels |
|---|---|---|---|---|
| 1Res | 11,702,717 | DownBlock2D | UpBlock2D | 256 |
| 1Res+1Attn | 13,391,549 | CrossAttnDownBlock2D, DownBlock2D | UpBlock2D, CrossAttnUpBlock2D | (128, 256) |
| 2Attn | 18,519,485 | CrossAttnDownBlock2D, CrossAttnDownBlock2D | CrossAttnUpBlock2D, CrossAttnUpBlock2D | (128, 256) |
| 2Res | 11,810,621 | DownBlock2D, DownBlock2D | UpBlock2D, UpBlock2D | (128, 256) |

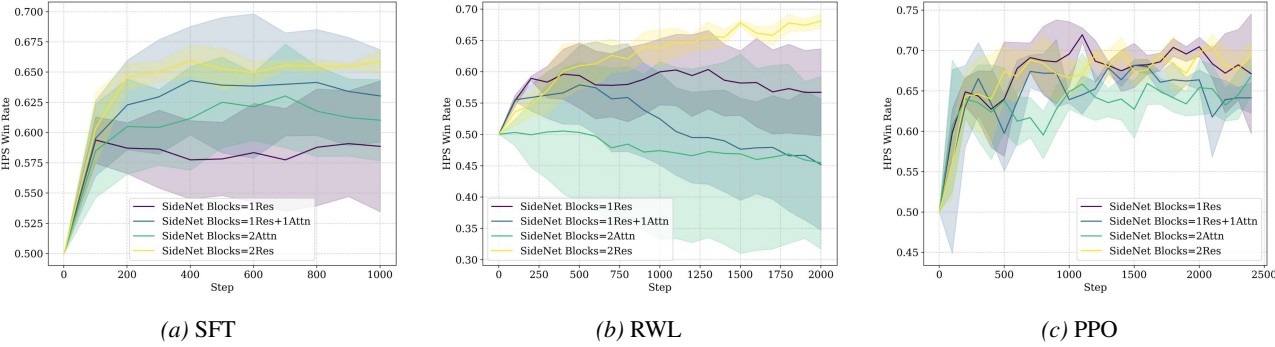

*(a)* SFT        *(b)* RWL        *(c)* PPO

*Figure 13.* Ablation studies on DiffCon block structures evaluated by the HPS-v2 win rate curve. Each plot contains four curves corresponding to 1Res, 1Res+1Attn, 2Attn, and 2Res (main-paper setting).

**Ablation studies on the input of the side network.** Under our parameterization (27), we input $\mu_0(x_t, c, t)$ instead of $x_t$ to the side network. We found this leads to better performance for RWL, see Figure 14.

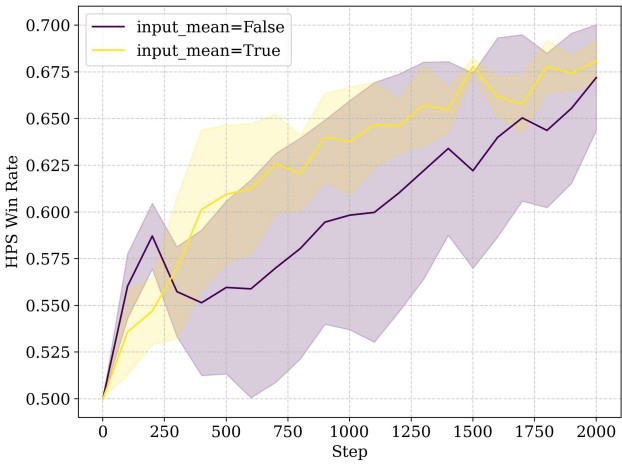

*Figure 14.* Ablation studies on the input of DiffCon for RWL, evaluated by the HPS-v2 win rate curve. The plot contains two curves corresponding to input_mean=True or False, i.e., using the mean $\mu_0(x_t, c, t)$ or $x_t$ as the input to the side network $s_\theta$.

