}\left(x_{t+1}|x_t, c\right) = p_{0,t}\left(x_{t+1}|x_t, c\right)\frac{Z_{t+1}\left(x_{t+1}, c\right)}{Z_t\left(x_t, c\right)}. \tag{57}$$

Let $\mathbb{P}_{u^\star}(x_{1:T}|c)$ be the joint distribution of $x_{1:T}$ under the optimal control $u^\star$. Then by (57), we have

$$
\begin{aligned}
\mathbb{P}_{u^\star}(x_{1:T}|c) &= p_{u^\star,-1}(x_0|c)\prod_{t=0}^{T-1}\mathbb{P}_{u^\star,t}\left(x_{t+1}|x_t, c\right) \\
&= \prod_{t=0}^{T-1} p_{0,t}\left(x_{t+1}|x_t, c\right)\frac{Z_{t+1}\left(x_{t+1}, c\right)}{Z_t\left(x_t, c\right)} \\