# OpenReview forum: "Diffusion Controller: Framework, Algorithms and Parameterization"
_ICML.cc/2026/Conference — ICML 2026 regular_

### Official Review · Reviewer_JVRS · 2026-02-18

**Soundness:** 3
**Presentation:** 3
**Significance:** 3
**Originality:** 3
**Overall Recommendation:** 4
**Confidence:** 4

**Summary:**

This paper proposes Diffusion Controller (DiffCon), a unified control-theoretic view of diffusion fine-tuning that formulates reverse diffusion sampling as state-only stochastic control in (generalized) linearly-solvable MDPs. Control is defined as reweighting the pretrained reverse-time transition kernels with an $f$-divergence regularizer. The paper derives practical RL fine-tuning objectives including an $f$-regularized policy-gradient with a PPO-style update and a reward-weighted regression objective with a minimizer-preservation statement under KL. It also motivates a parameterization where the learned score decomposes into a frozen pretrained baseline plus a lightweight controller side-network conditioned on exposed intermediate denoising outputs for gray-box access. The paper reports improvements on Stable Diffusion v1.4 for supervised and reward-driven alignment tasks.

**Compliance With Llm Reviewing Policy:**

Affirmed.

**Key Questions For Authors:**

see the weaknesses above

**Limitations:**

yes

**Strengths And Weaknesses:**

Strengths

* The LS-MDP framing is a clean way to unify several diffusion RLFT objectives, and the derivations connect policy-gradient style updates and reward-weighted regression under a single regularized-control lens (e.g., $D_f$ control cost).

* The pretrained-plus-controller decomposition (frozen backbone + side network) is a reasonable design for gray-box settings and is supported by empirical wins in SFT/RWL.



Weaknesses

1. The paper positions DiffCon as unifying “patchwork heuristics,” but there is already a sizable body connecting diffusion fine-tuning to entropy or divergence-regularized stochastic control and RLFT variants. The difference here seems to be (i) emphasizing LS-MDP  rather than introducing explicit actions, and (ii) the specific parameterization based on a controller correction. I think the paper should be more explicit about what is new beyond existing entropy/KL-regularized RL diffusion fine-tuning, especially since the derived updates resemble known forms when $D_f=D_{\mathrm{KL}}$ and/or when $\omega=0$.

2. Some theoretical statements feel “template-like” and depends on choices that are not well motivated. The controlled kernel is defined as
$P_{u,t}(x_{t+1}\mid x_t,c)=p_{0,t}(x_{t+1}\mid x_t,c)\exp(u_t(x_{t+1},x_t,c))$ with normalization, and the objective is
$r_t(x_t,c)+\mathbb{E}[V_{t+1}]-\omega D_f(P_{u,t}(\cdot\mid x_t,c),|,p_{0,t}(\cdot\mid x_t,c))$.
This is elegant, but many conclusions hinge on the exact regularizer $f$ and the coefficient $\omega$. In practice, $\omega$ is a main tuning knob, yet the paper mostly treats it as a given. I would expect either giudance on selecting $\omega$ (or sensitivity plots in the main paper), or at least a stronger arguement that the main benefits are robust across a reasonable range.


3. Math clarity issue: the reward-weighted regression objective is clean under KL, but for general $f$ it becomes less precise. Theorem 1 defines
$L_f(\rho)=\frac12\mathbb{E}\big[w_f(r(x_T,c),\omega),|\epsilon-\rho(\sqrt{\bar\alpha_t}x_T+\sqrt{1-\bar\alpha_t}\epsilon,c,t)|2^2\big]$
with weights like $w{\mathrm{KL}}(r,\omega)=\exp(r/\omega)$ for KL. But the paper also says that for $D_f\neq D_{\mathrm{KL}}$ the induced marginal $p_\omega^*$ typically differs from $p_{u_\omega}$, and there is generally no closed-form $w_f$ exactly matching the original target. That weaken the “same minimizer” story in the general $f$ case, which is exactly when the framwork is advertised as generalized. I think the paper should be more direct: either focus on $D_{\mathrm{KL}}$ as the practically meaningful case, or provide a clearer approximation guaranteed for non-KL divergences.


4. The gray-box claim depends on what signals are actually exposed. The parameterization uses a side network conditioned on intermediate outputs such as the implied reverse mean $\mu_0(x_t,c,t)$, and the score is written roughlyas
$\rho_\omega(x_t,c,t)=\rho_0(x_t,c,t)+\zeta_{\mathrm{model}},s_\omega(\mu_0(x_t,c,t),c,t)$.
This is reasonable if the API exposes $\rho_0$ or $\mu_0$, but in many “black-box-ish” deployments you might only get final images (or limited guidance knobs), not per-step noise predictions. So, the setting is closer to “partial white-box” than true gray-box in some realistic scenarios. This should be clarified as an assumption about access.

5. The empirical results are mixed when compared to LoRA, especially for PPO. In the main results table, the gray-box DiffCon score under PPO is much lower than LoRA (about 0.6957 vs. 0.9048 win rate against pretrained), while the joint and separate white-box variants (DiffCon-J and DiffCon-S) slightly outperform LoRA. This is important because a key claim is that the gray-box method can match or beat LoRA, which seems true for SFT and RWL but not for PPO in the reported checkpoint. The paper should acknowlege this limitation instead of suggesting uniform superiority.

6. Evaluation relies on a single reward model and win-rate metric. The main automatic evaluation is HPS-v2 win rate, with some sanity checks. But reward-model overfitting is a known issue in RLFT. I would like to see additional evaluation that is less coupled to the same preference model, for example human eval for more settings (not only PPO), or cross-reward generalization (optimize with HPS-v2 but evaluate with other preference scorers) to check whether the gains are real alignment improvements rather than reward hacking.

---

> ### Author Rebuttal · Authors · 2026-03-31
>
> # Response to Reviewer JVRS
>
> Thank you for your valuable feedback! If our responses below resolve the remaining questions, we would be grateful for your consideration of raising the score; we would also welcome any questions.
>
> >W1: regarding introduction
>
> We agree. We will better position DiffCon relative to prior entropy/KL-regularized diffusion control and clarify that our novelty is not the existence of a control connection itself, but the discrete-time LS-MDP formulation, direct reverse-kernel view, and resulting parameterization/algorithms. We will also soften the “patchwork heuristics” wording.
>
> >W2: regarding hyperparameter $\tau$
>
> We will clarify that the coefficient ($\omega$ in your comment; $\tau$ in our notation) is the reward-vs.-fidelity knob induced by the divergence penalty: too small over-optimizes reward and can destabilize training, while too large is overly conservative. We do already provide supporting ablations: Fig. 10 shows PPO performs best at $\tau_{KL}=10^{-4}$, while too small/zero KL hurts reward; Fig. 11 shows RWL performs best at $\tau_{RWL}=5\times10^{-4}$, and lines 1644–1645 note smaller values are unstable. We will make this motivation and tuning guidance more explicit. We will also clarify that the choice of $f$ is meaningful both theoretically and empirically: Theorem 1 induces different weighting rules (e.g., exponential for KL, polynomial for $\alpha$), and in RWL the polynomial variant used in the main results (Figure 2, middle) outperforms the exponential one (Figure 12).
>
> >W3: regarding general $f$-divergence
>
> We agree that the reward-weighted regression story is sharpest under KL, and we did make this distinction explicit below Theorem 1: for general $f$-divergences, (19) should be interpreted as a principled surrogate associated with the $f$-regularized optimal marginal $\widetilde{p}^\star$, which need not equal $p_{u^\star}$.
>
> We nevertheless keep the general $f$-divergence for three reasons. (i) The core DiffCon framework broader than KL: Eq. (9), Proposition 1 and PPO-style update all apply to arbitrary $f$-divergences. (ii) Even outside KL, Eq. (19) remains meaningful because it optimizes score matching under. (iii) Empirically, we do not study KL alone: our RWL setup uses the polynomial $\alpha$-divergence weighting in Eq. (21), and App. D also reports the exponential KL weighting in Eq. (20).
>
> We appreciate the suggestion and will revise the paper to sharpen this KL-versus-general-$f$ distinction.
>
> >W4: regarding the gray-box assumption
>
> We agree that the strict final-output-only black-box setting is outside the scope of DiffCon. If only final images (or a few guidance knobs) are exposed, the relevant problem is outer-loop optimization around a fixed API—e.g., RL/prompt expansion—rather than controlling or finetuning the diffusion reverse process itself. Our intended regime is a limited-interface frozen-backbone setting: the pretrained denoiser remains sealed and unmodified, but intermediate denoising signals are exposed by the interface. Regarding terminology, we use “gray-box” to distinguish this regime from white-box finetuning setting that requires much stronger assumption that the pretrained backbone can be directly modified. We'll clarify this scope more explicitly in the paper.
>
> >W5: regarding PPO results
>
> We agree that for PPO, LoRA outperforms gray-box DiffCon in Table 1. Our claim in Sec. 5.2 is not that gray-box DiffCon uniformly outperforms LoRA; rather, the main comparison is within each access regime: gray-box DiffCon vs. DiffCon-Naive, and white-box DiffCon-J/S vs. LoRA. The stronger cross-setting result applies only to SFT and RWL, where gray-box DiffCon can surpass white-box LoRA despite the more restrictive access setting. Since LoRA is not deployable in the gray-box regime, that cross-setting comparison is a bonus rather than the primary criterion. That said, we agree that the abstract/introduction could read too broadly, and we will revise them to make this distinction explicit, especially for PPO.
>
> >W6: regarding evaluation
>
> Beyond HPS-v2 win rate, Appendix D reports CLIP, CLIP-Aesthetics, and PickScore for SFT/RWL/PPO (Tables 8–10); these remain broadly stable as HPS-v2 improves, which argues against simple overfitting to HPS-v2; we also include qualitative samples for SFT/RWL/PPO (Tables 5–7), which do not show the kind of visibly degenerate images typically associated with reward hacking. The paper already reports human evaluation for PPO (Fig. 3d), and during rebuttal we additionally ran human evaluation for SFT and RWL on 50 prompts from the HPS-v2 prompt set:
> | Human eval win rate vs. Pretrained | DiffCon-J | DiffCon-S | DiffCon |
> |---|---:|---:|---:|
> | SFT | 0.552 | 0.656 | 0.593 |
> | RWL | 0.545 | 0.559 | 0.516 |
> These results suggest the gains are not solely an artifact of overfitting to HPS-v2.

---

> > ### Author Rebuttal · Reviewer_JVRS · 2026-04-02
> >
> > Thank you for the detailed and thoughtful rebuttal. I appreciate your wilingness to engage with the feedback and clarify the scope of your work.
> >
> > Here is my feedback based on your responses:
> >
> > * **Positioning & Terminology (W1 & W4):** I am glad you will soften the "patchwork heuristics" language and more clearly delineate your work from existing entropy-regularized diffusion control literature. Clarifying the exact requirements for your "gray-box" setting (needing access to intermediate denoising signals rather than just a final API) is crucial for setting reader expectasons.
> > * **Hyperparameters & $\tau$ (W2):** It is helpfull that you will make the tuning guidance for $\tau$ more explicit in the main text, drawing on the ablations already present in the appendixe.
> > * **General $f$-divergences (W3):** I appreciate your defense of keeping the general $f$-divergence framing. As long as the distinction between the exact KL case and the surrogate nature of the general $f$-divergence case for reward-weighted regression is made very clear in the main text, this is acceptable.
> > * **PPO Results (W5):** Thank you for agreeing to nuance the abstract and introduction regarding the PPO results. It is important to accurately reflect that while gray-box DiffCon is competitive in SFT and RWL, it does not match white-box LoRA in the PPO setting.
> > * **Evaluation (W6):** The new human evaluation results for SFT and RWL are a strong addition and help mitigate concerns about reward hacking. Please ensure these are included in the final version.
> >
> > The rebuttal successfully clarifies several ambiguities and improves the framing of the paper's contributions. The LS-MDP perspective is an elegant way to unify these algorithms, and the proposed parameterization is a practical contribution for settings where the backbone must remain frozen. However, the core theoretical and empirical contributions, while solid, are somewhat incremental over existing regularized diffusion fine-tuning works.
> >
> > Therefore, I will maintain my score.

---

### Official Review · Reviewer_ryn4 · 2026-02-24

**Soundness:** 3
**Presentation:** 3
**Significance:** 3
**Originality:** 3
**Overall Recommendation:** 4
**Confidence:** 4

**Summary:**

**The submission's central area pertains to** the unification of controllable diffusion generation through a control-theoretic lens. **This article proceeds to focus on an important theme**: casting reverse diffusion sampling as a state-only stochastic control problem within generalized Linearly-Solvable Markov Decision Processes (LS-MDPs). By deriving optimality conditions from $f$-divergence regularized objectives, the authors propose a principled score decomposition. This motivates a "gray-box" parameterization—**DiffCon**—which utilizes a side network conditioned on intermediate denoising outputs ($\mu_0$) to steer the pretrained model without modifying its frozen backbone.

**Compliance With Llm Reviewing Policy:**

Affirmed.

**Final Justification:**

I recommend acceptance of this paper.

The work is theoretically sound, presenting a clear and principled formulation of controllable diffusion through the LS-MDP framework. The proposed score decomposition and DiffCon parameterization are well-motivated and grounded in the derived optimality conditions. In terms of originality, the paper offers a novel and elegant unification of several diffusion fine-tuning methods under a control-theoretic perspective. Empirically, it shows meaningful gains over strong gray-box baselines.

My main concerns in the original review were about (1) comparison fairness with LoRA, (2) dependence on intermediate states, and (3) clarity on latent vs. pixel space. The rebuttal successfully addressed my misunderstanding regarding LoRA, provided additional experimental evidence, and fixed presentation issues (e.g., title inconsistency). This increased my confidence in the empirical validity of the method.

Some limitations remain (e.g., inference overhead and deployment constraints), but these are reasonable trade-offs rather than fundamental flaws.

Overall, the paper makes a novel, well-justified, and empirically supported contribution.

**Key Questions For Authors:**

### **Problems for Rebuttal**

#### **1. Inference Parameters vs. Methodological Efficacy**
LoRA adds zero parameters to the inference pipeline after merging. DiffCon, however, requires an active 12M-parameter side network.
*   **Question**: Is the performance gain a result of the LS-MDP formulation, or simply a byproduct of the increased total parameter capacity during inference? Would a LoRA with an equivalent inference compute budget (e.g., much higher rank) still be outperformed by DiffCon?

#### **2. Dependency on Intermediate Latent States**
In most enterprise-grade API scenarios, intermediate denoising trajectories are strictly hidden.
*   **Question**: Since DiffCon requires accessing and modifying the latent $\mu_0$ at every step, how can this "gray-box" method function in a true "black-box" SaaS environment? Is there a version of DiffCon that works with only terminal image feedback without per-step intervention?

#### **3. Theoretical Clarity on Latent vs. Pixel Space**
The paper uses the variable $x$ in its mathematical proofs, which typically implies pixel space, but Stable Diffusion operates in a latent space $z$.
*   **Question**: Does the "Score Decomposition" derived in the paper hold the same semantic meaning in high-resolution pixel spaces where noise patterns are less structured than in VAE latents? The authors should clarify if DiffCon was tested on non-latent models (e.g., pixel-space DDPM) to verify if the control correction remains as effective without the VAE's latent prior.

**Limitations:**

yes

**Strengths And Weaknesses:**

### **Strengths**
1.  **Elegant Theoretical Unification**: The paper provides a rigorous mathematical bridge between diffusion models and LS-MDPs. This framework elegantly unifies various heuristic fine-tuning methods (like PPO and reward-weighted regression) into a single control-theoretic view.
2.  **Principled Architecture**: The design of the side network is not arbitrary; it is directly derived from the optimality conditions of the regularized control problem. Using $\mu_0$ (the implied clean image) as a guidance signal is a theoretically grounded choice.
3.  **Strong Empirical Results**: DiffCon demonstrates consistent gains over LoRA and other gray-box baselines in HPS-v2 win rates across SFT and RLFT settings, showing superior parameter efficiency in terms of training.

### **Weaknesses**
1.  **Inference Overhead and Mergeability**: Unlike LoRA, which can be merged into the backbone weights for zero-cost inference, DiffCon requires the side network to run in parallel during every denoising step. This introduces persistent computational and memory overhead during sampling.
2.  **Practical SaaS Deployment Limitations**: The reliance on intermediate latent states ($\mu_0$) at each timestep $t$ is a major bottleneck. Current black-box SaaS APIs (e.g., DALL-E 3) do not expose these trajectories, rendering the method unusable in such ubiquitous environments.
3.  **Conceptual Ambiguity between Spaces**: While the theory is presented using $x$-space notation (typical for pixel-space diffusion), the experiments are conducted on Stable Diffusion (Latent Space). The paper lacks a clear distinction or proof of whether this control logic transfers seamlessly to pure pixel-space diffusion (e.g., standard DDPM) without the semantic compression of a VAE latent space. This inconsistency in notation vs. implementation creates a gap in the theoretical generalizability.

*Minor Issue: Title Inconsistency*
I noticed a discrepancy between the title in the submitted PDF ("Diffusion Controller: Framework, Parameterization, and Algorithms") and the title recorded in the submission system. While this does not impact the technical evaluation of the work, the authors should ensure consistent metadata for final publication.

---

> ### Author Rebuttal · Authors · 2026-03-31
>
> # Response to Reviewer ryn4
>
> Thank you for your positive and thoughtful review! We hope our responses below clarify the remaining questions. If so, we would greatly appreciate your consideration in raising your score; we would also be very grateful for any further questions.
>
> > W1+Q1: regarding inference overhead, mergeability and methodological efficacy
>
> Thank you for this point. We agree that in the white-box setting, LoRA can be merged in to the backbone, but this does not mean the adaptation has zero inference cost: as shown in Table 1, LoRA contains $1.7\times 10^7$ parameters, more than any of our proposed parameterization methods ($1.2\times 10^7$ for DiffCon, $1.6\times 10^7$ for DiffCon-J and DiffCon-S). These additional parameters add to the inference cost, and mergeability alone does not determine inference cost. To quantify the inference efficiency, we measured inference time on 500 HPD-v2 prompts: Pretrained=148.18 seconds, DiffCon = 183.97s, DiffCon-Naive = 181.68s, LoRA = 221.41s, DiffCon-S = 213.03s, and DiffCon-J = 229.94s. These results demonstrate (i) lora does not have "zero-cost" inference; (ii) our white-box methods are comparable to LoRA, and our gray-box methods are faster than LoRA.
>
> Therefore, regarding your question: *"Is the gain just from more inference-time parameters?"*
> the answer is **No**. First, DiffCon does not have more parameters than LoRA. Second, the clearest evidence that the gain comes from the method rather than merely attaching a side network is the DiffCon vs. DiffCon-Naive comparison: both methods use the same side-network backbone, but DiffCon applies the structured controller form (28), whereas DiffCon-Naive uses a direct residual correction. The substantial gap between them therefore points to the parameterization, not just the presence of an additional network.
> And for your question *"Would a LoRA with an equivalent inference compute budget still be outperformed by DiffCon?"* The answer is **Yes**, as our current evidence show we outperform LoRA with a similar/smaller inference time cost.
>
> We also want to stress that our comparison is not meant to deny LoRA’s mergeability; rather, it targets for the different gray-box regime  where the pretrained backbone cannot be modified and only limited denoising interfaces are exposed. In that regime, mergeable backbone adapters such as LoRA are not directly applicable, whereas DiffCon remains deployable.
>
> > W2+Q2: regarding dependency on intermediate latent states
>
> Thank you for this comment. We agree that strict output-only SaaS APIs are outside the scope of DiffCon. However, we believe the limitation "the reliance on intermediate latent states is a major bottleneck" is overstated. As formalized in Sec. 2, DiffCon targets for a gray-box regime, where the pretrained backbone is sealed but exposes limited denoising interfaces. This is a meaningful and increasingly common interface in prior work, see the related work given in Sec. 2 for example. Moreover, this assumption is substantially weaker than the white-box (model internal) access used by most diffusion RLFT baselines.
>
> In a strict black-box setting, adaptation may still be possible, but the problem changes: it becomes outer-loop optimization around a fixed API (e.g., prompt expansion or prompt-level RL), rather than fine-tuning or controlling the diffusion reverse process itself. Our method is designed for the latter limited-interface diffusion setting, and we will clarify this scope more explicitly in the paper.
>
> >W3+Q3: regarding the gap on latent v.s. pixel space
>
> Thank you for raising this. In our analysis, $x_t$ denotes the diffusion state of the backbone model. Thus, for latent diffusion models such as Stable Diffusion, $x_t$ should be interpreted as the latent variable $z_t$, not as an RGB image. With this interpretation, the LS-MDP derivation and the score decomposition carry over directly to latent diffusion by replacing the data distribution $p_{data}(x∣c)$ with the latent distribution induced by the VAE encoder, $p_{lat}(z|c)$, and replacing $x_t$ by $z_t$ throughout. The final semantic effect is then realized in image space through the fixed VAE decoder.
>
> Empirically, our current instantiation is the latent-space version where we fix the VAE. However extending DiffCon to joint VAE + latent-diffusion finetuning is straightforward and does not discard the latent prior. One keeps the same latent-space controller on $z_t$, while allowing the VAE to receive a small residual update. This VAE update only refines the pretrained VAE slightly when beneficial, and does not remove its prior. This is fully consistent with our parameterization, which treats the controlled score as a correction to a pretrained baseline and already supports joint residual adaptation in DiffCon-J.
>
> >(minor) title inconsistency
>
> We will correct the title to "Latent Diffusion Controller: Framework, Algorithms and Parameterization". Thank you for noticing this!

---

> > ### Author Rebuttal · Reviewer_ryn4 · 2026-04-01
> >
> > Thanks for authors' rebuttal, I previously held some misconceptions regarding LoRA; the author has clearly addressed the inaccuracies in my original review comments and has also provided the runtime comparison tests I requested—which is excellent. However, given that I have already assigned this paper a very high score, I do not believe that raising it further would substantially increase the likelihood of acceptance. Therefore, I have decided to maintain my original score.

---

### Official Review · Reviewer_P4VY · 2026-03-10

**Soundness:** 3
**Presentation:** 3
**Significance:** 2
**Originality:** 2
**Overall Recommendation:** 4
**Confidence:** 3

**Summary:**

This paper introduces Diffusion Controller (DiffCon), a framework that reinterprets diffusion fine‑tuning as a linearly solvable MDP, where the reverse process is controlled by reweighting the pretrained dynamics under an f-divergence regularizer. This perspective yields a unified derivation of policy gradient, PPO, and reward‑weighted losses. The optimal control solution further motivates a lightweight side‑network that conditions on the pretrained mean and requires only a frozen backbone—enabling effective gray‑box fine‑tuning. Experiments on Stable Diffusion v1.4 with HPSv2 show that DiffCon improves human‑preference alignment across supervised, reward‑weighted, and PPO settings, often outperforming white‑box baselines like LoRA with fewer trainable parameters.

**Compliance With Llm Reviewing Policy:**

Affirmed.

**Final Justification:**

Since the authors have addressed my previous concerns, I raised my rating from Weak Reject to Weak Accept.

**Key Questions For Authors:**

1. Would you solve the concern mentioned in the first point of weaknesses?
2. Could you provide more experimental results on more baselines/diffusion models with related methods (DDPO, DPOK) to demonstrate the effectiveness of the proposed method?
3. Could you quantify the added cost of the side network during training or inference, parameter counts and training time against baselines and other related methods?
The above three issues are my primary concerns. If they can be definitively resolved, I will increase my rating.

**Limitations:**

This paper does not appear to provide limitations; I have listed what I consider to be limitations under “weaknesses”.

**Strengths And Weaknesses:**

Strengths:
1. Casting diffusion fine‑tuning as an LS‑MDP provides a clean, unified perspective that naturally yields policy gradient, PPO, and reward‑weighted losses; the optimal control analysis also guides the network design.
2. The side network conditioned on the pretrained mean requires only a frozen backbone and a lightweight module, making it compatible with proprietary or safety‑sealed models while outperforming full fine‑tuning in some settings.
3. The LS‑MDP formulation works with any f-divergence and can be combined with different learning signals (paired data, reward models, online RL), offering flexibility for various adaptation tasks.

Weaknesses:
1. The core prerequisite for DiffCon’s gray-box adaptation is access to the backbone’s intermediate denoising outputs. If the backbone only provides per-timestep noisy samples with no query access to the pretrained score or reverse mean, the side network’s required input cannot be analytically computed, invalidating DiffCon’s core parameterization and preventing direct implementation. Forcibly applying the side network here only reduces it to the paper’s DiffCon-Naive baseline (directly taking noisy samples as input), which experiments show performs far worse than DiffCon, forfeiting its core advantages — fully aligning with the "less straightforward" conclusion.
2. While DiffCon often outperforms LoRA in SFT and RWL, the improvements are sometimes modest, and the PPO results, though high, are not directly compared against other PPO‑based diffusion fine‑tuning methods (e.g., DDPO).
3. The paper does not quantify the added cost of the side network during training or inference, nor does it compare parameter counts and training time against baselines, leaving efficiency claims only qualitative.

---

> ### Author Rebuttal · Authors · 2026-03-30
>
> # Response to Reviewer P4VY
>
> Thank you for your valuable feedback. We hope this response addresses your key concerns. If it does, we would be grateful if you would consider updating your score. We would welcome any additional questions.
>
> > W1/Q1: regarding the gray-box assumption
>
> Thank you for raising this. We acknowledge that our gray-box setting is not $x_t$-only: as stated in the paper, DiffCon assumes a frozen backbone that exposes intermediate denoising outputs--such as per-step noise predictions or the implied reverse mean $\mu_0$-while keeping the backbone weights inaccessible. Although this is not the most minimalist possible API, it is a common and reasonable interface in the literature, see the "Access levels for diffusion finetuning." paragraph in Section 2 for related work. Under this gray-box setting, access to $\mu_0$ provides the controller with a more informative denoising signal than raw $x_t$ alone, enabling DiffCon to exploit more backbone information while keeping the backbone intact.
>
> More importantly, the value of our gray-box DiffCon does not depend solely on the availability of intermediate denoising outputs. Even if we replace the side-network input $\mu_0$ by $x_t$ in (27), the method does not collapse into DiffCon-Naive.
> DiffCon-Naive differs not only in input choice, but also in parameterization: DiffCon employs the structured side network $s_\theta$ as specified in (28), whereas DiffCon-Naive directly adds an unrestricted side-network output to the pretrained score. Thus, a DiffCon-style controller fed with the same input $x_t$ remains different from DiffCon-Naive.
>
> Empirically, this distinction is supported by our ablations: Fig. 14 already shows for RWL that using $x_t$ as side-network input is only slightly worse than using the reverse mean. And by comparing Fig. 14 with Fig. 2 we can see that our DiffCon with input $x_t$ still substantially outperforms DiffCon-Naive under RWL.
> We will add the analogous SFT and PPO plots to the appendix, both of which show very similar performance between the two input choices, indicating that the reverse-mean input is helpful but not a prerequisite for DiffCon’s gains. Below we provide a few datapoints from the omitted plots:
>
> **SFT**
>
> | Step | DiffCon w/ $x_t$ (`input_mean=False`) | DiffCon w/ $\mu_0(x_t,c,t)$ (`input_mean=True`) | Gap $(\mu_0 - x_t)$ |
> |---:|---:|---:|---:|
> | 100  | 0.595 | 0.603 | +0.008 |
> | 200  | 0.631 | 0.645 | +0.014 |
> | 400  | 0.649 | 0.658 | +0.009 |
> | 700  | 0.663 | 0.660 | -0.003 |
> | 1000 | 0.668 | 0.667 | -0.001 |
>
> **PPO**
>
> | Step | DiffCon w/ $x_t$ (`input_mean=False`) | DiffCon w/ $\mu_0(x_t,c,t)$ (`input_mean=True`) | Gap $(\mu_0 - x_t)$ |
> |---:|---:|---:|---:|
> | 500  | 0.658 | 0.673 | +0.015 |
> | 1000 | 0.685 | 0.671 | -0.014 |
> | 1500 | 0.684 | 0.675 | -0.009 |
> | 2000 | 0.693 | 0.699 | +0.006 |
> | 2400 | 0.686 | 0.696 | +0.010 |
>
> > W2/Q2: regarding PPO
>
> We agree that for PPO, the performance gains over LoRA are modest. This is likely attributable to the high baseline performance of LoRA, which achieves 0.9048 HPS-v2 win rate (Table 1), leaving limited headroom for further improvement. For this reason, we conducted human evaluation (Figure 3(d)) to provide additional validation.
>
> We also want to stress that the current version already provides a theoretical and empirical comparison to PPO-based diffusion finetuning: in Section 3.2.1 (also Appendix A) we show that our policy-gradient formulation recovers DDPO in the unregularized case $\tau_{KL}=0$, and is closely related to DPOK. Fig. 10 ablates $\tau_{KL}\in${$0,10^{-5},10^{-4},5\times 10^{-4}, 10^{-3}$}, so the $\tau_{KL}=0$ setting corresponds to the DDPO realization, and hence Fig. 10 demonstrates DDPO fails to improve reward, supporting the benefit of our regularization.
>
> > W3/Q3: regarding efficiency quantification
>
> Thank you for this question. We do already report parameter counts in Table 1: DiffCon and DiffCon-Naive use $1.2\times 10^7$ trainable parameters, DiffCon-J and DiffCon-S use $1.6\times 10^7$, while the LoRA uses $1.7\times 10^7$. Thus, all of our methods use fewer trainable parameters than LoRA. Also note that the added controller is very small compared to the pretrained SD v1.4 (which has ~1B parameters).
>
> For the rebuttal, we also measured inference time on the same 500 prompts: DiffCon = 183.97s, DiffCon-Naive = 181.68s, LoRA = 221.41s, DiffCon-S = 213.03s, and DiffCon-J = 229.94s. These results indicate that both gray-box and white-box setting have essentially the same inference cost as their baselines, and the gray-box methods yield faster inference than the white-box methods.
>
> Regarding training time, in our experiments, SFT and RWL training times are similar across methods and much smaller than PPO training times. For PPO the gray-box settings are more efficient than the white-box ones: the measured time per 100 steps was 02:32:08 for LoRA, 02:35:04 for DiffCon-J, 00:55:40 for DiffCon-Naive, and 00:55:52 for DiffCon.

---

> > ### Author Rebuttal · Reviewer_P4VY · 2026-04-03
> >
> > Since the authors have addressed my previous concerns, I raised my rating from Weak Reject to Weak Accept.

---

### Official Review · Reviewer_YtmU · 2026-03-16

**Soundness:** 3
**Presentation:** 3
**Significance:** 3
**Originality:** 4
**Overall Recommendation:** 4
**Confidence:** 4

**Summary:**

This paper studies controllable generation in diffusion models and proposes diffusion controller (DiffCon), a control-theoretic framework that formulates the reverse diffusion process as a state-only stochastic control problem within a linearly-solvable MDP. Simulation demonstrates better quality-efficiency tradeoff compared with baseline methods.

**Compliance With Llm Reviewing Policy:**

Affirmed.

**Key Questions For Authors:**

1. The evaluation can be improved. The evaluation mainly focuses on text-to-image preference alignment, and other diffusion control tasks can also be added. The simulation is limited to a single backbone model.
2. The computational complexity and efficiency of the proposed method can be discussed.
3. Pictures can be added to illustrate the proposed method.

**Limitations:**

No, the authors should discuss the limitations and potential negative societal impact of this work.

**Strengths And Weaknesses:**

1. This paper provides a clear theoretical formulation and learning objectives. However, the evaluation mainly focuses on text-to-image preference alignment, and other diffusion control tasks can also be added. The simulation is limited to a single backbone model. The computational complexity and efficiency of the proposed method can be discussed.
2. This paper is well structured. However, more pictures and explanations can be added to illustrate the proposed method.
3. The controllable generation and alignment are important.
4. This work looks novel.

---

> ### Author Rebuttal · Authors · 2026-03-30
>
> # Response to Reviewer YtmU
>
> Thank you for your positive review and constructive feedback! If our response below resolve the remaining questions, we would greatly appreciate your consideration in updating your score. And we are more than happy to answer your further questions.
>
> >Q1: regarding the evaluation
>
> Thank you for this helpful suggestion! We agree that evaluating on broader tasks and backbone models would further strengthen the paper, and we also highlighted extending DiffCon to other settings as promising future direction in the conclusion. For the current submission, we chose to focus the empirical study on text-to-image alignment as a representative and practically important testbed that lets us isolate the main contribution under both supervised and reward-driven fine-tuning. Even within this scope, the experiments already cover multiple fine-tuning algorithms (including SFT, RWL, and PPO), compare several parameterization methods (LoRA,(gray-box) DiffCon, DiffCon-Naive, DiffCon-J, DiffCon-S), and include additional appendix results and ablations. We agree, however, that broader task and backbone coverage would be valuable, and we will add this limitation more explicitly and expand the evaluation in a revised version where feasible. Thank you for your advice!
>
> >Q2: regarding complexity and efficiency
>
> Thank you for this question. We agree that the computational complexity and efficiency of the proposed method should be stated more explicitly. From a computational standpoint, DiffCon does not change the number of diffusion sampling steps, and it does not introduce any inner-loop optimization during sampling. Its per-step cost is the cost of the pretrained denoiser plus a lightweight controller network, so the overall inference complexity remains the same order as the baseline diffusion sampler, i.e., linear in the number of denoising steps, with only a small constant-factor overhead from the controller. This is also consistent with our design motivation: the controller is a small correction module added on top of a frozen pretrained backbone rather than a second full diffusion model.
>
> More importantly, in terms of parameter efficiency, Table 1 shows that (gray-box) DiffCon and DiffCon-Naive use $1.2\times 10^7$ trainable parameters, DiffCon-J and DiffCon-S use $1.6\times 10^7$, while the LoRA baseline uses $1.7\times 10^7$. Thus, all of our proposed variants use fewer trainable parameters than LoRA, supporting our claim that our methods are more parameter-efficient while achieving higher performance. Also note that the added controller is very small compared to the pretrained SD v1.4 (which has ~1B parameters).
>
> To further quantify runtime efficiency, we additionally measured inference time on the same 500 prompts: DiffCon = 183.97s, DiffCon-Naive = 181.68s, LoRA = 221.41s, DiffCon-S = 213.03s, and DiffCon-J = 229.94s. These results suggest that both gray-box and white-box setting have essentially the same inference cost as their baselines, and the gray-box methods yield faster inference than the white-box methods.
>
> We will clarify these complexity and efficiency points in the revised paper. Thank you for your comment.
>
> >Q3: Pictures can be added to illustrate the proposed method.
>
> Thank you for raising this point. We agree that additional visualizations and explanation would help readers better understand the proposed method. In response, we will add a dedicated overview figure that summarizes the proposed DiffCon pipeline: it will illustrate (i) the pretrained reverse diffusion process as the passive dynamics, (ii) how DiffCon reweights these transitions under the LS-MDP formulation to balance the terminal objective and the divergence regularization, and (iii) how this leads in practice to a frozen pretrained backbone plus a lightweight side controller conditioned on intermediate denoising outputs. We will also expand the method section with more intuitive, step-by-step explanation of how the control-theoretic formulation connects to the two practical training objectives in the paper, namely the policy-gradient/PPO update and the reward-weighted regression objective.
>
> >Limitations: missing limitation discussion
>
> Thank you for raising this. We will add the following paragraph to our paper:
>
> **Limitations and societal impact.** Our current evaluation is limited to text-to-image alignment on a single backbone, Stable Diffusion v1.4. Although the LS-MDP formulation and controller parameterization are intended to apply more broadly to diffusion control, validation in other settings—such as personalization, safety steering, and transfer or domain adaptation—and on additional backbones remains future work. More broadly, while DiffCon can improve alignment and controllability, it is inherently dual-use: biased feedback or unsafe objectives may over-optimize proxies or facilitate harmful content. Careful reward design, safety filtering, and responsible deployment are therefore essential.

---

### Decision · Program_Chairs · 2026-04-30

**Decision:**

Accept (regular)

**Comment:**

This paper introduces DiffCon, a framework that casts reverse diffusion sampling as a state-only stochastic control problem within linearly solvable MDPs. The approach formulates control by reweighting pretrained reverse-time transition kernels using an f-divergence cost. This formulation leads to practical reinforcement learning fine-tuning methods, including an f-divergence-regularized policy-gradient and a reward-weighted regression objective. The authors implement a side-network parameterization conditioned on intermediate denoising outputs, allowing adaptation while keeping the backbone frozen.

Reviewers found the control-theoretic MDP formulation elegant and appreciated the mathematically grounded. Primary reviewer concerns included the evaluation being limited to a single backbone model, questions regarding the reliance on intermediate denoising outputs for "gray-box" deployments where application programming interfaces might be restricted, and missing quantitative measurements for computational and inference overhead compared to baselines like LoRA. The authors adequately addressed the core concerns during the rebuttal period. They provided inference time and parameter count comparisons demonstrating that DiffCon is more parameter-efficient than LoRA. The authors also provided new tabular data showing the method remains robust even when restricted to noisy samples $x_t$ rather than relying strictly on the full reverse-mean.